# Adversarial Training of Self-supervised Monocular Depth Estimation against Physical-World Attacks

**Zhiyuan Cheng**
Purdue University
cheng443@purdue.edu

**James Liang**
Rochester Institute of Technology
jcl3689@rit.edu

**Guanhong Tao**
Purdue University
taog@purdue.edu

**Dongfang Liu**[*]
Rochester Institute of Technology
dongfang.liu@rit.edu

**Xiangyu Zhang**[*]
Purdue University
xyzhang@cs.purdue.edu

## ABSTRACT

Monocular Depth Estimation (MDE) is a critical component in applications such as autonomous driving. There are various attacks against MDE networks. These attacks, especially the physical ones, pose a great threat to the security of such systems. Traditional adversarial training method requires ground-truth labels hence cannot be directly applied to self-supervised MDE that does not have ground-truth depth. Some self-supervised model hardening techniques (*e.g.*, contrastive learning) ignore the domain knowledge of MDE and can hardly achieve optimal performance. In this work, we propose a novel adversarial training method for self-supervised MDE models based on view synthesis without using ground-truth depth. We improve adversarial robustness against physical-world attacks using $L_0$-norm-bounded perturbation in training. We compare our method with supervised learning based and contrastive learning based methods that are tailored for MDE. Results on two representative MDE networks show that we achieve better robustness against various adversarial attacks with nearly no benign performance degradation.

## 1 INTRODUCTION

Monocular Depth Estimation (MDE) is a technique that estimates depth from a single image. It enables 2D-to-3D projection by predicting the depth value for each pixel in a 2D image and serves as a very affordable replacement for the expensive Lidar sensors. It hence has a wide range of applications such as autonomous driving (Liu et al., 2021a), visual SLAM (Wimbauer et al., 2021), and visual relocalization (Liu et al., 2021b), etc. In particular, self-supervised MDE gains fast-growing popularity in the industry (*e.g.*, Tesla Autopilot (Karpathy, 2020)) because it does not require the ground-truth depth collected by Lidar during training while achieving comparable accuracy with supervised training. Exploiting vulnerabilities of deep neural networks, multiple digital-world (Zhang et al., 2020; Wong et al., 2020) and physical-world attacks (Cheng et al., 2022) against MDE have been proposed. They mainly use optimization-based methods to generate adversarial examples to fool the MDE network. Due to the importance and broad usage of self-supervised MDE, these adversarial attacks have posed a great threat to the security of applications such as autonomous driving, which makes the defense and MDE model hardening an urgent need.

Adversarial training (Goodfellow et al., 2014) is the most popular and effective way to defend adversarial attacks. However, it usually requires ground truth labels in training, making it not directly applicable to self-supervised MDE models with no depth ground truth. Although contrastive learning gains a lot of attention recently and has been used for self-supervised adversarial training (Ho & Nvasconcelos, 2020; Kim et al., 2020), it does not consider the domain knowledge of depth estimation and can hardly achieve optimal results (shown in Section 4.2). In addition, many existing adversarial training methods do not consider certain properties of physical-world attacks such as strong perturbations. Hence in this paper, we focus on addressing the problem of *hardening self-supervised MDE models against physical-world attacks without requiring the ground-truth depth.*

---

[*]Corresponding authors.

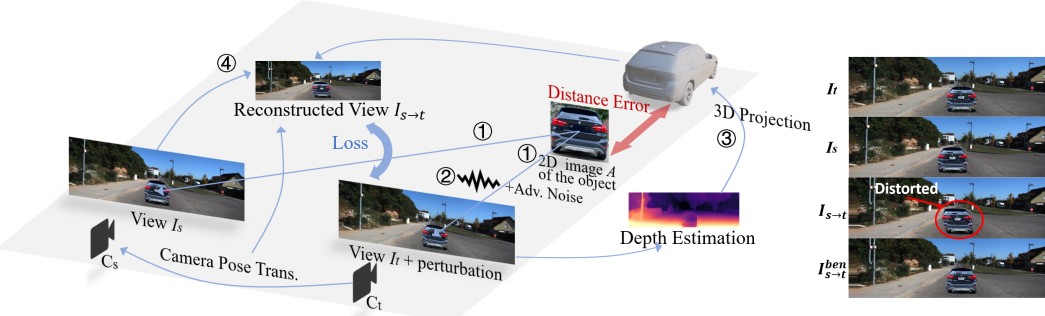

(a) Conceptual illustration.

(b) Different Views.

Figure 1: Self-supervised adversarial training of MDE with view synthesis.

A straightforward proposal to harden MDE models is to perturb 3D objects in various scenes and ensure the estimated depths remain correct. However, it is difficult to realize such adversarial training. First, 3D perturbations are difficult to achieve in the physical world. While one can train the model in simulation, such training needs to be supported by a high-fidelity simulator and a powerful scene rendering engine that can precisely project 3D perturbations to 2D variations. Second, since self-supervised MDE training does not have the ground-truth depth, even if realistic 3D perturbations could be achieved and used in training, the model may converge on incorrect (but robust) depth estimations. In this paper, we propose a new self-supervised adversarial training method for MDE models. Figure 1a provides a conceptual illustration of our technique. A board $A$ printed with the 2D image of a 3D object (*e.g.*, a car) is placed at a fixed location (next to the car at the top-right corner). We use two cameras (close to each other at the bottom) $C_t$ and $C_s$ to provide a stereo view of the board (images $I_t$ and $I_s$ in Figure 1b). Observe that there are fixed geometric relations between pixels in the two 2D views produced by the two respective cameras such that the image in one view can be transformed to yield the image from the other view. Intuitively, $I_t$ can be acquired by shifting $I_s$ to the right. Note that when two cameras are not available, one can use two close-by frames in a video stream to form the two views as well. During adversarial training, camera $C_t$ takes a picture $I_t$ of the original 2D image board $A$. Similarly, camera $C_s$ takes a picture $I_s$ of the board too (step ❶). The bounding box of the board $A$ is recognized in $I_t$ and the pixels in the bounding box corresponding to $A$ are perturbed (step ❷). Note that these are 2D perturbations similar to those in traditional adversarial training. At step ❸, the perturbed image $I_t$+perturbations is fed to the MDE model to make depth estimation, achieving a 3D projection of the object. Due to the perturbations, a vulnerable model generates distance errors as denoted by the red arrow between $A$ and the projected 3D object in Figure 1a. At step ❹, we try to reconstruct $I_t$ from $I_s$. The reconstruction is parameterized on the cameras' relative pose transformations and the estimated distance of the object from the camera. Due to the distance error, the reconstructed image $I_{s \to t}$ (shown in Figure 1b) is different from $I_t$. Observe that part of the car (the upper part inside the red circle) is distorted. In comparison, Figure 1b also shows the reconstructed image $I_{s \to t}^{ben}$ without the perturbation, which is much more similar to $I_t$. The goal of our training (of the subject MDE model) is hence to reduce the differences between original and reconstructed images. The above process is conceptual, whose faithful realization entails a substantial physical-world overhead. In Section 3, we describe how to avoid the majority of the physical-world cost through image synthesis and training on synthesized data.

While traditional adversarial training assumes bounded perturbations in $L_2$ or $L_\infty$ norm (i.e., measuring the overall perturbation magnitude on all pixels), physical-world attacks are usually unbounded in those norms. They tend to be stronger attacks in order to be persistent with environmental condition variations. To harden MDE models against such attacks, we utilize a loss function that can effectively approximate the $L_0$ norm (measuring the number of perturbed pixels regardless of their perturbation magnitude) while remaining differentiable. Adversarial samples generated by minimizing this loss can effectively mimic physical attacks. We make the following contributions:

① We develop a new method to synthesize 2D images that follow physical-world constraints (e.g., relative camera positions) and directly perturb such images in adversarial training. The physical world cost is hence minimized. ② Our method utilizes *the reconstruction consistency* from one view to the other view to enable self-supervised adversarial training without the ground-truth depth labels. ③ We generate $L_0$-bounded perturbations with a differentiable loss and randomize the camera and object settings during synthesis to effectively mimic physical-world attacks and improve robustness.

④ We evaluate the method and compare it with a supervised learning baseline and a contrastive learning baseline, adapted from state-of-the-art adversarial contrastive learning (Kim et al., 2020). Results show that our method achieves better robustness against various adversarial attacks with nearly no model performance degradation. The average depth estimation error of an adversarial object with 1/10 area of perturbation is reduced from 6.08 m to 0.25 m by our method, better than 1.18 m by the supervised learning baseline. Moreover, the contrastive learning baseline degrades model performance a lot. A demo is available at `https://youtu.be/_b7E4yUFB-g`.

## 2    RELATED WORKS

**Self-supervised MDE.** Due to the advantage of training without the depth ground truth, self-supervised MDE has gained much attention recently. In such training, stereo image pairs and/or monocular videos are used as the training input. Basically, two images taken by camera(s) from adjacent poses are used in each optimization iteration. A depth network and a pose network are used to estimate the depth map of one image and the transformation between the two camera poses, respectively. With the depth map and pose transformation, it further calculates the pixel correspondence across the images and then tries to rearrange the pixels in one image to reconstruct the other. The pose network and the depth network are updated simultaneously to minimize the reconstruction error. Garg et al. (2016) first propose to use color consistency loss between stereo images in training. Zou et al. (2018) enable video-based training with two networks (one depth network and one pose network). Many following works improve the self-supervision with new loss terms (Godard et al., 2017; Bian et al., 2019; Wang et al., 2018; Yin & Shi, 2018; Ramamonjisoa et al., 2021; Yang et al., 2020) or include temporal information (Wang et al., 2019; Zou et al., 2020; Tiwari et al., 2020; Watson et al., 2021). Among them, Monodepth2 (Godard et al., 2019) significantly improves the performance with several novel designs such as minimum photometric loss selection, masking out static pixels, and multi-scale depth estimation. Depthhints (Watson et al., 2019) further improves it via additional depth suggestions obtained from stereo algorithms. While such unsupervised training is effective, how to improve its robustness against physical attack remains an open problem.

**MDE Attack and Defense.** Mathew et al. (2020) use a deep feature annihilation loss to launch perturbation attack and patch attack. Zhang et al. (2020) design a universal attack with a multi-task strategy and Wong et al. (2020) generate targeted adversarial perturbation on images which can alter the depth map arbitrarily. Hu & Okatani (2019) propose a defense method against perturbation attacks by masking out non-salient pixels. It requires another saliency prediction network. For physical-world attacks, Cheng et al. (2022) generates a printable adversarial patch to make the vehicle disappear. To the best of our knowledge, we are the first work focusing on improving the robustness of self-supervised MDE models against physical-world attacks.

**Adversarial Robustness.** It is known that deep neural networks (DNN) are vulnerable to adversarial attacks. Imperceptible input perturbations could lead to model misbehavior (Szegedy et al., 2013; Madry et al., 2018; Moosavi-Dezfooli et al., 2016). Typically, adversarial training is used to improve the robustness of DNN models. It uses both benign and adversarial examples for training (Madry et al., 2018; Carlini & Wagner, 2017; Tramèr et al., 2018). Adversarial training has been applied to many domains like image classification (Carlini & Wagner, 2017; Madry et al., 2018), object detection (Zhang & Wang, 2019; Chen et al., 2021b;a), and segmentation (Xu et al., 2021; Hung et al., 2018; Arnab et al., 2018) etc. A common requirement for adversarial training is supervision because generating adversarial examples needs ground truth, and most tasks require labels for training. Some semi-supervised adversarial learning methods (Carmon et al., 2019; Alayrac et al., 2019) use a small portion of labeled data to enhance robustness. Contrastive learning (Ho & Nvasconcelos, 2020; Kim et al., 2020) is also used with adversarial examples either for better self-supervised learning or to improve robustness. In this work, we explore the adversarial training of MDE without using ground-truth depth and compare our method with contrastive learning-based and supervised learning-based methods that are specifically tailored for MDE. There are other defense techniques such as input transformations. However, Athalye et al. (2018a) point out that these techniques largely rely on obfuscated gradients which may not lead to true robustness. In the scenarios of autonomous driving, there are other works focusing on the security of Lidar or multi-sensor fusion-based systems (Tu et al., 2020; 2021; Cao et al., 2019; 2021). They use sensor spoofing or adversarial shapes to fool the Lidar hardware or AI model, while in this work, we consider fully-vision-based autonomous driving systems in which MDE is the key component.

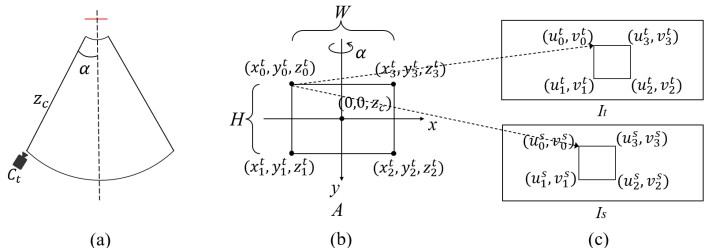

Figure 2: (a) Bird view of the relative positions of the camera and the target object. (b) 3D coordinates of the four corners of the object in the camera frame. (c) Projection from the physical-world object to the two views.

## 3 OUR DESIGN

Our technique consists of a few components. The first one (Section 3.1) is view synthesis that generates the two views $I_t$ and $I_s$ of the object and is equivalent to step ❶ in Figure 1a. The second one (Section 3.2) is robust adversarial perturbation that perturbs $I_t$ to induce maximum distance errors (step ❷ in Figure 1a). The third one (Section 3.3) is the self-supervised adversarial training (steps ❸ and ❹). Each of the contributing components is detailed below.

### 3.1 VIEW SYNTHESIS TO AVOID PHYSICAL WORLD SCENE MUTATION

As mentioned in Section 1, conceptually we need to place an image board of the 3D object at some physical locations on streets and use two cameras to take pictures of the board. In order to achieve robustness in training, we ought to vary the image of the object (e.g., different cars), the position of the image board, the positions and angles of the cameras, and the street view. This entails enormous overhead in the physical world. Therefore, we propose a novel view synthesis technique that requires minimal physical world overhead. Instead, it directly synthesizes $I_s$ and $I_t$ with an object in some street view, reflecting different settings of the aforementioned configurations.

Specifically, we use one camera to take a picture $I_t$ of a 2D image board $A$ of the object, with the physical width and height $W$ and $H$, respectively. As shown in Figure 2 (b), the four corners of the board have the 3D coordinates $(x_0^t, y_0^t, z_0^t)$, ..., and $(x_3^t, y_3^t, z_3^t)$, respectively, in the camera frame, namely, the coordinate system with the camera as the origin. The camera is placed at $z_c$ distance away from the board with an angle of $\alpha$, as shown in Figure 2 (a). The size of the board is true to the rear of the object. This is important for the later realistic synthesis step.

After that, we derive a projection function that can map a pixel in $A$ to a pixel in $I_t$. The function is parameterized on $W$, $H$, $z_c$, $\alpha$, etc. such that *we can directly use it to synthesize a large number of $I_t$'s with different $A$'s by changing those parameter settings.* To acquire $I_s$ that is supposed to form a stereo view with $I_t$, we do not necessarily need another camera in the physical world. Instead, we can use two neighboring video frames of some street view (*e.g.*, from the KITTI dataset (Geiger et al., 2013)), denoted as $R_t$ and $R_s$, to approximate a stereo pair taken by two close-by cameras. Note that the differences between the two images implicitly encode the relative positions of the two cameras. A prominent benefit of such approximation is that a large number of camera placements and street views can be easily approximated by selecting various neighboring video frames. This is consistent with existing works (Godard et al., 2019; Watson et al., 2019). We replace the area in $I_t$ that does not correspond to $A$, i.e., the background of the object, with the corresponding area in $R_t$. Intuitively, we acquire a realistic $I_t$ by stamping the object's image to a background image. The synthesis respects physical world constraints. A projection function parameterized by $R_t$ and $R_s$ can be derived to map a pixel in one camera's view to a pixel in the other. Then, we project the part of $I_t$ that corresponds to $A$ using the projection function and stamp it on $R_s$, acquiring $I_s$. As such, the resulted view of $A$ (in $I_s$) is consistent with the camera pose denoted by $R_s$. $I_t$ and $I_s$ are then used in model hardening (discussed later in Section 3.3).

Formally, if the center of the physical camera's view aligns with the center of the image board $A$, the correlation between a pixel $(u^A, v^A)$ in $A$ and its 3D coordinate $(x^t, y^t, z^t)$ is:

$$\begin{bmatrix} x^t \\ y^t \\ z^t \\ 1 \end{bmatrix} = \begin{bmatrix} \cos\alpha & 0 & -\sin\alpha & 0 \\ 0 & 1 & 0 & 0 \\ \sin\alpha & 0 & \cos\alpha & z_c \\ 0 & 0 & 0 & 1 \end{bmatrix} \cdot \begin{bmatrix} W/w & 0 & -W/2 \\ 0 & H/h & -H/2 \\ 0 & 0 & 0 \\ 0 & 0 & 1 \end{bmatrix} \cdot \begin{bmatrix} u^A \\ v^A \\ 1 \end{bmatrix}, \tag{1}$$

where $w$ and $h$ are the width and height of $A$ in pixels. The other variables ($e.g.$, $\alpha$, $z_c$, $W$, $H$, $etc.$) are defined in Figure 2. The 3D coordinates can be further projected to pixels in $I_t$ and $I_s$ as:

$$
\begin{aligned}
\left[u^t\ v^t\ 1\right]^\top &= 1/z^t \cdot K \cdot \left[x^t\ y^t\ z^t\ 1\right]^\top, \\
\left[u^s\ v^s\ 1\right]^\top &= 1/z^s \cdot K \cdot \left[x^s\ y^s\ z^s\ 1\right]^\top, \quad \left[x^s\ y^s\ z^s\ 1\right]^\top = T_{t\to s} \cdot \left[x^t\ y^t\ z^t\ 1\right]^\top,
\end{aligned}
\tag{2}
$$

where $T_{t\to s}$ is *the camera pose transformation* (CPT) that projects 3D coordinates in the physical camera $C_t$'s coordinate system to coordinates in the other (virtual) camera $C_s$'s coordinate system. It is determined by $R_s$ and $R_t$ as mentioned before. $K$ is the camera intrinsic. Combining Equation 1 and Equation 2, we know the projections from pixel $(u^A, v^A)$ of the object image to pixel $(u^t, v^t)$ in $I_t$ and to pixel $(u^s, v^s)$ in $I_s$. Let $\left[u^t\ v^t\ 1\right]^\top = P^{A\to t}_{z_c,\alpha}(u^A, v^A)$ and $\left[u^s\ v^s\ 1\right]^\top = P^{A\to s}_{z_c,\alpha,T_{t\to s}}(u^A, v^A)$. We synthesize $I_t$ and $I_s$ as:

$$
I_t[u,v] = \begin{cases} A[u^A, v^A], & [u\ v\ 1]^\top = P^{A\to t}_{z_c,\alpha}(u^A, v^A) \\ R_t[u,v], & otherwise \end{cases},
\tag{3}
$$

$$
I_s[u,v] = \begin{cases} A[u^A, v^A], & [u\ v\ 1]^\top = P^{A\to s}_{z_c,\alpha,T_{t\to s}}(u^A, v^A) \\ R_s[u,v], & otherwise \end{cases},
\tag{4}
$$

where $R_t$ and $R_s$ are the background images implicitly encoding the camera relative poses. A large number of $I_t$ and $I_s$ are synthesized by varying $R_t$, $R_s$, $z_c$, $\alpha$, $A$, and used in hardening. The creation induces almost zero cost compared to creating a physical world dataset with similar diversity.

## 3.2 Robust Adversarial Perturbations

We use an optimization based method to generate robust adversarial perturbations $\delta$ on the object image $A$ composing the corresponding adversarial object $A + \delta$ and synthesize $I'_t$ by replacing $A$ with $A + \delta$ in Equation 3. The synthesized $I'_t$ is then used in adversarial training. We bound the perturbations with $L_0$-norm, which is to constrain the number of perturbed pixels. Compared with digital-world attacks that use traditional $L_\infty$-norm or $L_2$-norm-bounded perturbations ($e.g.$, FGSM (Goodfellow et al., 2014), Deepfool (Moosavi-Dezfooli et al., 2016), and PGD (Madry et al., 2018)), physical-world attacks usually use adversarial patches (Brown et al., 2017) without restrictions on the magnitude of perturbations in the patch area because stronger perturbations are needed to induce persistent model misbehavior in the presence of varying environmental conditions ($e.g.$, lighting conditions, viewing angles, distance and camera noise). Hence, $L_0$-norm is more suitable in physical-world attacks because it restricts the number of pixels to perturb without bounding the perturbation magnitude of individual pixels. However, the calculation of $L_0$-norm is not differentiable by definition and hence not amenable for optimization. We hence use a soft version of it as proposed in Tao et al. (2022). The main idea is to decompose the perturbations into positive and negative components and use the long-tail effects of $\tanh$ function, in the normalization term, to model the two ends of a pixel's value change ($i.e.$, zero perturbation or arbitrarily large perturbation). As such, a pixel tends to have very large perturbation or no perturbation at all.

$$
\delta = maxp \cdot (\texttt{clip}(b_p, 0, 1) - \texttt{clip}(b_n, 0, 1)).
\tag{5}
$$

$$
\mathcal{L}_{pixel} = \sum_{h,w} \left( \max_c \left( \frac{1}{2}(\tanh(\frac{b_p}{\gamma}) + 1) \right) \right) + \sum_{h,w} \left( \max_c \left( \frac{1}{2}(\tanh(\frac{b_n}{\gamma}) + 1) \right) \right).
\tag{6}
$$

Specifically, the perturbation is defined in Equation 5 and the normalization term is $\mathcal{L}_{pixel}$ in Equation 6, where $b_p$ and $b_n$ are the positive and negative components; $\texttt{clip()}$ bounds the variable to a range of [0,1]; $h, w, c$ are the height, width and channels of image and $\gamma$ is a scaling factor. We refer readers to Tao et al. (2022) for detailed explanations.

Equation 7 formulates our perturbation generation, where $S_p$ is a distribution of physical-world distance and view angles (e.g., reflecting the relations between cameras and cars during real-world driving); $S_R$ is the set of background scenes (e.g., various street views); $D()$ is the MDE model which outputs the estimated depth map of given scenario and $MSE()$ is the mean square error.

$$
\min_{b_n, b_p}\ E_{z_c,\alpha \sim S_p, R_t \sim S_R} \left[ MSE \left( D\left(I'_t\right)^{-1}, 0 \right) \right] + \mathcal{L}_{pixel}, \quad s.t.\ L_0(\delta) \le \epsilon.
\tag{7}
$$

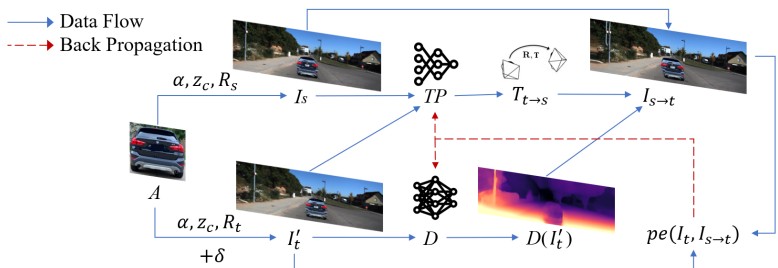

Figure 3: Pipeline of adversarial training of self-supervised monocular depth estimation. Solid lines denote data flow and dashed lines denote back propagation paths.

Our adversarial goal is to make the target object further away, so we want to maximize the depth estimation (*i.e.*, minimize the reciprocal). Intuitively, we synthesize $I'_t$ with random $R_t$ and different $\alpha$ and $z_c$ of object $A$ and use expectation of transformations (EoT) (Athalye et al., 2018b) to improve physical-world robustness. We minimize the mean square error between zero and the reciprocal of synthesized scenario's depth estimation in the adversarial loss term and use $\mathcal{L}_{pixel}$ as the normalization term of perturbations. Parameter $\epsilon$ is a predefined $L_0$-norm threshold of perturbations and it denotes the maximum ratio of pixels allowed to perturb (*e.g.*, $\epsilon = 1/10$ means $1/10$ pixels can be perturbed at most).

### 3.3 SELF-SUPERVISED MDE TRAINING.

In each training iteration, we first synthesize $I_t$ and $I_s$ as mentioned earlier. Perturbations are then generated to $I_t$ to acquire $I'_t$ following Equation 7. As illustrated in Figure 1a, we reconstruct a version of $I_t$ from $I_s$ using the depth information derived from $I'_t$. We call the resulted image $I_{s \to t}$. Intuitively, $I'_t$ causes depth errors which distort the projection from $I_s$ to $I_{s \to t}$ such that the latter appears different from $I_t$. Similar to how we project $(u^A, v^A)$ to $(u^t, v^t)$ or $(u^s, v^s)$ earlier in this section (i.e., Equation 2), we can project $(u^s, v^s)$ in $I_s$ to a pixel $(u^{s \to t}, v^{s \to t})$ in $I_{s \to t}$. This time, we use the depth information derived from $I'_t$. Formally, the projection is defined as:

$$\begin{bmatrix} x^{s \to t} & y^{s \to t} & z^{s \to t} & 1 \end{bmatrix}^\top = K^{-1} \cdot D_{I'_t}(u^{s \to t}, v^{s \to t}) \cdot \begin{bmatrix} u^{s \to t} & v^{s \to t} & 1 \end{bmatrix}^\top,$$
$$\begin{bmatrix} x^s & y^s & z^s & 1 \end{bmatrix}^\top = T_{t \to s} \cdot \begin{bmatrix} x^{s \to t} & y^{s \to t} & z^{s \to t} & 1 \end{bmatrix}^\top, \quad \begin{bmatrix} u^s & v^s & 1 \end{bmatrix}^\top = 1/z^s \cdot K \cdot \begin{bmatrix} x^s & y^s & z^s & 1 \end{bmatrix}^\top. \tag{8}$$

Intuitively, there are relations between 2D image pixels and 3D coordinates, i.e., the first and the third formulas in Equation 8. The 3D coordinates also have correlations decided by camera poses, i.e., the second formula. Observe that, the first 2D-to-3D relation is parameterized on $D_{I'_t}$, the depth estimation of $I'_t$. Let $\begin{bmatrix} u^s & v^s \end{bmatrix}^\top = P_{D_{I'_t}, T_{t \to s}}(u^{s \to t}, v^{s \to t})$ be the transformation function that projects a pixel in $I_{s \to t}$ to a pixel in $I_s$ derived from Equation 8. $I_{s \to t}$ is synthesized as:

$$I_{s \to t}[u, v] = I_s[P_{D_{I'_t}, T_{t \to s}}(u, v)]. \tag{9}$$

Intuitively, it rearranges the pixels in $I_s$ to form $I_{s \to t}$. We then compare $I_{s \to t}$ with $I_t$ and minimize their difference to guide the training.

Our training pipeline is shown in Figure 3. There are two trainable networks, the MDE model $D$ and a camera transposing model *TP*. Recall that we need the camera pose transformation matrix $T_{t \to s}$ between the two cameras' coordinate systems. We hence train the *TP* network that predicts $T_{t \to s}$ from a given pair of background images $R_s$ and $R_t$. We denote it as: $T_{t \to s} = TP(R_t, R_s)$. Observe that in Figure 3, from left to right, the pipeline takes the object image $A$ and synthesizes images $I_t$ and $I_s$. $I_t$ and $A$ are further used to derive adversarial sample $I'_t$, which is fed to the depth network $D$ to acquire depth estimation. The depth information, the *TP* network's output $T_{t \to s}$, and $I_s$ are used to derive $I_{s \to t}$. Two outputs $I_{s \to t}$ and $I_t$ are compared. The training objective is hence as:

$$\min_{\theta_D, \theta_{TP}} \mathcal{L}_p = pe(I_t, I_{s \to t}), \tag{10}$$

which is to update the weight values of $D$ and *TP* to minimize the photometric reconstruction error of the two outputs, denoted by $pe()$. Specific designs of $pe()$ differ in literature but our model hardening technique is general to all self-supervised MDE methods.

Table 1: **Benign performance** of original and hardened models on depth estimation.

| Models | Monodepth2 | | | | | DepthHints | | | | |
|---|---|---|---|---|---|---|---|---|---|---|
| | ABSE↓ | RMSE↓ | ABSR↓ | SQR↓ | δ↑ | ABSE↓ | RMSE↓ | ABSR↓ | SQR↓ | δ↑ |
| Original | 2.125 | 4.631 | 0.106 | 0.807 | 0.877 | 2.021 | 4.471 | 0.100 | 0.728 | 0.886 |
| L0+SelfSup (Ours) | 2.16 | 4.819 | 0.105 | 0.831 | 0.874 | 2.123 | 4.689 | 0.103 | 0.777 | 0.877 |
| L0+Sup | 2.162 | 4.648 | 0.110 | 0.846 | 0.876 | 2.015 | 4.453 | 0.100 | 0.734 | 0.887 |
| L0+Contras | 3.218 | 6.372 | 0.155 | 1.467 | 0.782 | 3.626 | 6.742 | 0.209 | 1.561 | 0.694 |
| PGD+SelfSup | 2.169 | 4.818 | 0.105 | 0.826 | 0.874 | 2.120 | 4.680 | 0.103 | 0.774 | 0.877 |
| PGD+Sup | 2.153 | 4.637 | 0.109 | 0.838 | 0.876 | 2.019 | 4.460 | 0.101 | 0.736 | 0.886 |
| PGD+Contras | 3.217 | 6.083 | 0.194 | 1.825 | 0.756 | 3.928 | 7.526 | 0.213 | 2.256 | 0.701 |

* For hardened models, A+B denotes generating adversarial perturbation with method A and training with method B.

## 4 EVALUATION

In this section, we evaluate the performance of our method in white-box, black-box, and physical-world attack scenarios, and discuss the ablations. Our code is available at `https://github.com/Bob-cheng/DepthModelHardening`.

### 4.1 EXPERIMENTAL SETUP.

**Networks and Dataset.** We use Monodepth2 (Godard et al., 2019) and DepthHints (Watson et al., 2019) as our subject networks to harden. They are representative and popular self-supervised MDE models that are widely used as benchmarks in the literature. Both models are trained on the KITTI dataset (Geiger et al., 2013) and our methods fine-tune the original models publicly available.

**Baselines.** There are no direct baselines available since no prior works have been focusing on hardening MDE models as far as we know. Hence we extend state-of-the-art contrastive learning-based and supervised learning-based adversarial training methods to MDE and use them as our baselines. They do not require ground-truth depth, same as our self-supervised method. Details are in Appendix A

**Training Setup.** In adversarial training, the ranges of distance $z_c$ and viewing angle $\alpha$ are sampled randomly from 5 to 10 meters and -30 to 30 degrees, respectively. The view synthesis uses EoT (Athalye et al., 2018b). We generate the adversarial perturbations with two methods: $L_0$-norm-bounded with $\epsilon = 1/10$ and $L_\infty$-norm-bounded (*i.e.*, PGD (Madry et al., 2018)) with $\epsilon = 0.1$. The latter is for comparison purposes. We train with our self-supervised method and two baseline methods based on contrastive learning and supervised learning. Hence, there are 6 approaches combining the 2 perturbation generation methods with the 3 training methods. With these approaches, we fine-tune the original model for 3 epochs on the KITTI dataset and produce 6 hardened models for each network. Other detailed configurations and the selection of 2D object images are in Appendix B.

**Attacks.** We conduct various kinds of attacks to evaluate the robustness of different models. They are $L_0$-norm-bounded attacks with $\epsilon = 1/20, 1/10, 1/5$ and $1/3$, $L_\infty$-norm-bounded (PGD) attacks with $\epsilon = 0.05, 0.1$ and $0.2$ (image data are normalized to [0,1]), and an adversarial patch attack in Mathew et al. (2020). Adversarial perturbation or patch is applied to an object image. The patch covers 1/10 of the object at the center. Each attack is evaluated with 100 randomly selected background scenes. The object is placed at a distance range of 5 to 30 meters and a viewing angle range of -30 to 30 degrees. We report the average attack performance over different background scenes, distances, and viewing angles for each attack. In addition, we conduct the state-of-the-art physical-world attack (Cheng et al., 2022) with the printed optimal patch and a real vehicle in driving scenarios. Adversarial examples are in Appendix D.

**Metrics.** We use the mean absolute error (ABSE), root mean square error (RMSE), relative absolute error (ABSR), relative square error (SQR), and the ratio of relative absolute error under 1.25 ($\delta$) as the evaluation metrics. These metrics are widely used in evaluating depth estimation performance. Metric $\delta$ denotes the percentage of pixels of which the ratio between the estimated depth and ground-truth depth is smaller than 1.25. It is the higher, the better and the others are the lower, the better. The definition of each metric can be found in Appendix C.

### 4.2 MAIN RESULTS

**Benign Performance.** Together with the original model, we have 7 models under test for each network. We evaluate the depth estimation performance on the KITTI dataset using the Eigen split and report the results in Table 1. As shown, self-supervised and supervised methods have little influence on the models' depth estimation performance, which means these approaches can harden the model with nearly no benign performance degradation. In contrast, the contrastive learning-

Table 2: **Defence performance** of original and hardened models under attacks.

| | Attacks | Original | | L0+SelfSup (Ours) | | L0+Sup | | L0+Contras | | PGD+SelfSup | | PGD+Sup | | PGD+Contras | |
|---|---|---|---|---|---|---|---|---|---|---|---|---|---|---|---|
| | | ABSE↓ | δ↑ | ABSE↓ | δ↑ | ABSE↓ | δ↑ | ABSE↓ | δ↑ | ABSE↓ | δ↑ | ABSE↓ | δ↑ | ABSE↓ | δ↑ |
| Monodepth2 | L0 1/20 | 4.71 | 0.65 | **0.18** | **0.99** | 0.44 | 0.98 | 1.30 | 0.67 | 0.49 | 0.95 | 0.58 | 0.94 | 0.69 | 0.94 |
| | L0 1/10 | 6.08 | 0.51 | **0.25** | **0.98** | 0.94 | 0.93 | 1.75 | 0.54 | 0.82 | 0.91 | 1.18 | 0.79 | 0.96 | 0.89 |
| | L0 1/5 | 8.83 | 0.39 | **0.34** | **0.9** | 1.59 | 0.85 | 2.32 | 0.46 | 2.33 | 0.70 | 2.72 | 0.51 | 1.11 | 0.85 |
| | L0 1/3 | 9.99 | 0.34 | **0.52** | **0.96** | 2.08 | 0.78 | 2.65 | 0.41 | 4.32 | 0.51 | 4.09 | 0.41 | 1.75 | 0.70 |
| | PGD 0.05 | 4.74 | 0.56 | 0.82 | 0.97 | 1.29 | 0.80 | 6.61 | 0.38 | **0.67** | **0.98** | 0.82 | 0.95 | 1.82 | 0.67 |
| | PGD 0.1 | 11.68 | 0.34 | 1.53 | 0.85 | 2.53 | 0.71 | 12.74 | 0.24 | **1.38** | **0.95** | 1.64 | 0.76 | 2.66 | 0.53 |
| | PGD 0.2 | 17.10 | 0.23 | **3.46** | **0.69** | 6.14 | 0.50 | 20.14 | 0.15 | 3.81 | 0.58 | 5.04 | 0.32 | 3.97 | 0.42 |
| | Patch | 2.71 | 0.77 | **0.39** | **0.98** | 1.35 | 0.89 | 6.40 | 0.52 | 0.40 | 0.98 | 0.84 | 0.92 | 0.50 | 0.95 |
| DepthHints | L0 1/20 | 2.33 | 0.66 | **0.19** | **0.99** | 0.34 | 0.96 | 1.06 | 0.83 | 0.22 | 0.99 | 0.58 | 0.89 | 0.40 | 0.99 |
| | L0 1/10 | 3.19 | 0.59 | **0.27** | **0.99** | 0.48 | 0.95 | 1.56 | 0.77 | 0.42 | 0.98 | 1.03 | 0.79 | 0.60 | 0.97 |
| | L0 1/5 | 4.77 | 0.42 | **0.40** | **0.98** | 0.96 | 0.82 | 1.85 | 0.75 | 0.83 | 0.92 | 1.93 | 0.68 | 0.66 | 0.95 |
| | L0 1/3 | 6.03 | 0.36 | **0.48** | **0.98** | 1.64 | 0.68 | 2.60 | 0.69 | 1.45 | 0.79 | 3.06 | 0.57 | 1.16 | 0.82 |
| | PGD 0.05 | 3.11 | 0.48 | **0.62** | **0.98** | 1.23 | 0.75 | 4.05 | 0.55 | 0.64 | 0.98 | 0.93 | 0.79 | 1.16 | 0.79 |
| | PGD 0.1 | 6.44 | 0.36 | 1.27 | 0.86 | 2.37 | 0.62 | 7.59 | 0.36 | **1.21** | **0.92** | 1.76 | 0.67 | 1.74 | 0.62 |
| | PGD 0.2 | 18.37 | 0.23 | 3.09 | 0.60 | 7.13 | 0.41 | 13.59 | 0.24 | 6.14 | **0.68** | 4.22 | 0.37 | **2.60** | 0.49 |
| | Patch | 0.70 | 0.91 | 0.46 | 0.95 | 0.53 | 0.93 | 6.90 | 0.49 | 0.46 | 0.95 | 0.36 | **0.99** | **0.34** | 0.98 |

*Bold and underlining indicate the best and second best performance in each row. Hardened models are named the same as Table 1.

based approach performs the worst. The ABSE of estimated depth is over 1 m worse than the original model. The reason could be that contrastive learning itself does not consider the specific task (*i.e.*, MDE) but fine-tunes the encoder to filter out the adversarial perturbations. Thus the benign performance is sacrificed during training. The other two methods consider the depth estimation performance either by preserving the geometric relationship of 3D space in synthesized frames or by supervising the training with estimated depth.

**White-box Attacks.** We conduct various white-box attacks on each model to evaluate the robustness of hardened models. Specifically, for each model, we compare the estimated depth of the adversarial scene (*i.e.*, $I'_t$) with that of the corresponding benign scene (*i.e.*, $I_t$ in Equation 3) and larger difference means worse defense performance. Table 2 shows the result. As shown, all the hardened models have better robustness than the original models against all kinds of attacks, and it is generic on the two representative MED networks, which validates the effectiveness of our adversarial training approaches. Comparing different approaches, L0+SelfSup has the best performance in general. It reduces the ABSE caused by all-level $L_0$-norm-bounded attacks from over 4.7 m to less than 0.6 m. Specifically, the self-supervision-based method outperforms the contrastive learning-based and the supervision-based methods regardless of the perturbation generation method used. It is because the self-supervision-based method follows the original training procedure that is carefully designed for the network and has been evaluated thoroughly. It is not surprising that models adversarially trained with $L_0$-norm-bounded perturbation (our method) achieve better robustness against $L_0$-norm-bounded attacks and so do PGD-based methods, but more importantly, $L_0$-norm-based training also has good defense performance against PGD attacks. The robustness of L0+SelfSup is only slightly worse than PGD+SelfSup on some PGD attacks and even better than it on stronger PGD attacks. An explanation is that $L_0$-norm does not restrict the magnitude of perturbation on each pixel, and stronger PGD attacks are closer to this setting (*i.e.*, high-magnitude perturbations) and can be well-defended using the $L_0$-norm-based adversarial training. Monodepth2 is vulnerable to the patch attack, and this kind of attack can be well-defended by our methods. L0+SelfSup also performs the best. Depthhints itself is relatively robust to the patch attack, and our methods can further reduce the attack effect. Our defense generalizes well to complex scenes including various road types, driving speed, and the density of surrounding objects. Qualitative results are in Appendix D.

**Black-box Attacks.** We also validate our methods against black-box attacks. We use the original Monodepth2 model and the models fine-tuned with $L_0$-norm-bounded perturbations and the three training methods. We perform $L_0$-norm-bounded attacks on each model with $\epsilon = 1/10$ and apply the generated adversarial object to other models evaluating the black-box

Table 3: Defence performance of original and hardened models under **black-box attacks**.

| Target | Original | | L0+SelfSup (Ours) | | L0+Sup | | L0+Contras | |
|---|---|---|---|---|---|---|---|---|
| Source | ABSE | δ↑ | ABSE | δ↑ | ABSE | δ↑ | ABSE | δ↑ |
| Original | - | - | **0.25** | **0.99** | 0.55 | 0.95 | 1.35 | 0.71 |
| L0+SelfSup | **0.52** | **0.93** | - | - | **0.24** | **0.98** | **0.45** | **0.95** |
| L0+Sup | 1.09 | 0.80 | **0.29** | **0.99** | - | - | 0.65 | 0.89 |
| L0+Contras | 2.65 | 0.50 | **0.22** | **0.99** | 0.27 | 0.98 | - | - |

*Bold indicates the best performance in each row or column.

attack performance. The first column in Table 3 denotes the source models and other columns are the target models' defense performance. Looking at each column, adversarial examples generated from L0+SelfSup have the worst attack performance, which indicates the low transferability of ad-

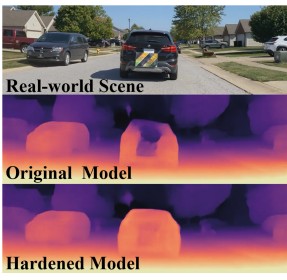

Figure 4: Physical-world attack and defence. Video: https://youtu.be/_b7E4yUFB-g

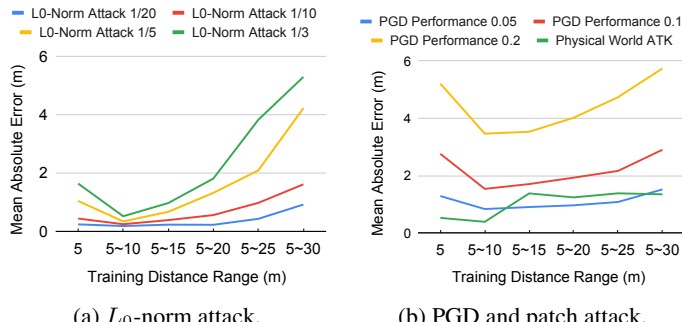

(a) $L_0$-norm attack.

(b) PGD and patch attack.

Figure 5: Robustness with different **training distance ranges**.

versarial examples generated from the model trained with our self-supervised methods. From each row, we can observe that `L0+SelfSup` has the best defense performance against adversarial examples generated from each source model, which further validates the robustness of `L0+SelfSup`. In summary, the self-supervised training method can produce safer and more robust models. Their adversarial examples have low transferability and they defend against black-box attacks well.

**Physical-world Attacks.** Our evaluation with the state-of-the-art physical-world MDE attack (Cheng et al., 2022) validates the effectiveness of our method in various real-world lighting conditions, driving operations, road types, etc. The experimental settings are the same as Cheng et al. (2022). Figure 4 shows the result. The first row is the real-world adversarial scene, in which a car is moving with an adversarial patch attached to the rear. The second row is the depth predicted by the original Monodepth2 model and the third row is predicted by our hardened model (`L0+SelfSup`). The "hole" in the output of the original model caused by the adversarial patch is fixed in our hardened model. It is known that the adversarial attacks designed for the physical world are still generated in the digital world and they have better digital-world attack performance than physical-world performance because additional environmental variations in the physical world degrade the effectiveness of adversarial patterns (Braunegg et al., 2020; Wu et al., 2020; Cheng et al., 2022). Hence, defending attacks in the digital world is more difficult and our success in digital-world defense in previous experiments has implied effectiveness in the physical world.

### 4.3 ABLATIONS

**Distance Range in Training.** While synthesizing views in training, the range of distance $z_c$ of the target object is randomly sampled from $d_1$ to $d_2$ meters. In this ablation study, we evaluate the effect of using different ranges of distance in training. We use `L0+SelfSup` to fine-tune the original Monodepth2 model. The ranges of distance we use in training are [5, 5], [5, 10], [5, 15], [5, 20], [5, 25] and [5, 30] (Unit: meter). Note that, for a fair comparison, the range of distance we use in model evaluation is always from 5 to 30 meters. The results are shown in Figure 5. As shown, the model trained with a distance range of 5-10 meters has the best robustness and a larger or smaller distance range could lead to worse performance. It is because further distances lead to smaller objects on the image and fewer pixels are affected by the attack. Thus the attack performance is worse at further distances and training with these adversarial examples is not the most effective. If the distance in training is too small (*e.g.*, 5 meters), the model cannot defend various scales of attack patterns and cannot generalize well to further distances. In our experiments, the range of 5-10 meters makes a good balance between training effectiveness and generality. Other ablation studies about transferability to unseen target objects is in Appendix E.

### 5 CONCLUSION

We tackle the problem of hardening self-supervised Monocular Depth Estimation (MDE) models against physical-world attacks without using the depth ground truth. We propose a self-supervised adversarial training method using view synthesis considering camera poses and use $L_0$-norm-bounded perturbation generation in training to improve physical-world attacks robustness. Compared with traditional supervised learning-based and contrastive learning-based methods, our method achieves better robustness against various adversarial attacks in both the digital world and the physical world with nearly no benign performance degradation.

## 6 REPRODUCIBILITY STATEMENT

To help readers reproduce our results, we have described the implementation details in our experimental setup (Section 4.1 and Appendix B). In the supplementary materials, we attached our source code and instructions to run. We will open our source code right after acceptance. The dataset we use is publicly available. We also attached the videos of our physical-world attack in the supplementary materials.

## 7 ACKNOWLEDGMENTS

This research was supported, in part by IARPA TrojAI W911NF-19-S-0012, NSF 2242243, 1901242 and 1910300, ONR N000141712045, N00014-1410468 and N000141712947.

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

# Appendix

## Adversarial Training of Self-supervised Monocular Depth Estimation against Physical-World Attacks

This document provides more details about our work and additional experimental settings and result. We organize the content of our appendix as follows:

- Section A: the baseline methods we tailored for MDE specifically.
- Section B: more details about the training configurations.
- Section C: the formal definition of the metrics we used in our evaluation.
- Section D: the adversarial attack examples and qualitative results of defense.
- Section E: the transferability evaluation of our hardened models to other target objects.
- Section F: comparing the supervised baseline trained with pseudo-depth labels and that trained with ground-truth depth labels.
- Section G: the broader impact and limitations.

## A    BASELINES

**Adversarial Contrastive Learning.** Contrastive learning is a widely used technique specifically tailored to self-supervised learning scenarios (Chen et al., 2020; He et al., 2020; Wu et al., 2018; Tian et al., 2020; Ye et al., 2019; Misra & Maaten, 2020). It has been used with adversarial examples either to improve the robustness of models against adversarial attacks (Kim et al., 2020) or to enhance the performance of contrastive learning itself (Ho & Nvasconcelos, 2020). In this work, we extend a state-of-the-art contrastive learning-based adversarial training method (Kim et al., 2020) to harden MDE models against physical attacks. We use it as a baseline to compare with our method.

Similar to Kim et al. (2020), the positive pairs in our contrastive learning are the benign examples ($I_t$) and the corresponding adversarial examples ($I'_t$), and we further augment those examples by changing the color. Different from Kim et al. (2020), we do not need negative pairs. Instead, we use a learning method proposed in SimSiam (Chen & He, 2021) that only requires positive pairs and can achieve competitive performance with smaller batch sizes. Figure 6 (a) shows the procedure. The key point is to maximize the similarity between the embeddings of the benign and adversarial examples so that their depth map outputs from the decoder network are similar. The parameters of the subject MDE model's encoder and the prediction MLP network are updated iteratively in training. We use color augmentation instead of other transformations (*e.g.*, resizing and rotation) because the embeddings should be similar among positive samples and the change of color would

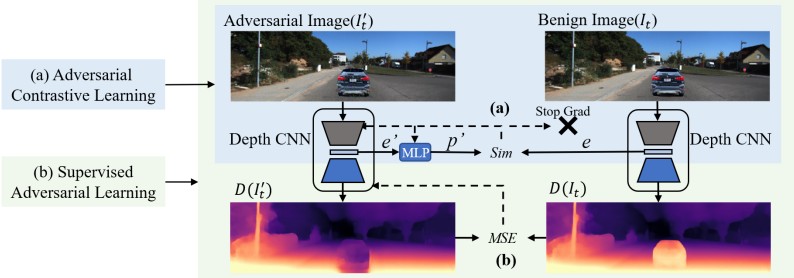

Figure 6: (a) Adversarial contrastive learning of model encoder. The color-augmented benign ($I_t$) and adversarial ($I'_t$) examples are fed to the depth model encoder (the grey block) and one embedding ($e'$) is then fed to a prediction multi-layer perceptron (MLP) that transcribes the embedding to another embedding $p'$. We maximize the similarity of the output ($p'$) with the other embedding ($e$). Backpropagation is only calculated along one side to update the encoder. (b) Supervised adversarial training with the estimated depth as the pseudo-ground truth. We use the output of the original model on benign examples as the ground truth to supervise the training of the subject model with adversarial examples. The solid lines denote data flow and the dashed lines denote back propagation paths.

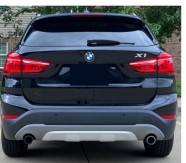 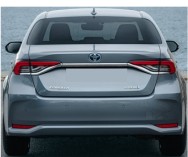 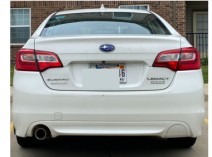 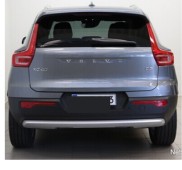 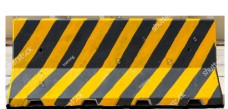

(a) BMW SUV Black (b) Toyota Sedan Blue (c) Subaru Sedan White (d) Volvo SUV Grey (e) Traffic Barrier

Figure 7: The various target objects in the transferability evaluation.

not affect the depth map output (but other transformations would). Other settings such as the MLP network structure are the same as SimSiam (Chen & He, 2021) and we refer readers to it for detailed explanations.

**Supervised Adversarial Learning with Estimated Depth.** Since we do not have depth ground truth in the self-supervised scenario, one alternative way to do adversarial training is to use the estimated depth by the original model with inputs of benign samples as the pseudo ground truth (i.e., pseudo labels) and perform supervised adversarial training. As shown in Figure 6 (b), we use mean square error (MSE) as the loss function to update MDE model parameters and minimize the difference between the model output of adversarial samples and the pseudo ground truth.

Using the pseudo ground truth predicted by an existing model is proved to be a simple and effective method in the field of semi-supervised learning (SSL) (Lee et al., 2013) and it has been used in adversarial training (Deng et al., 2021) and self-supervised MDE (Petrovai & Nedevschi, 2022) to boost model performance. Particularly, in the field of MDE, using pseudo-ground truth is good enough compared with using the real ground truth (Petrovai & Nedevschi, 2022). Same as our supervised baseline, Petrovai & Nedevschi (2022) uses the depth estimated by an existing MDE model (i.e., pseudo depth labels) to supervise the following MDE model training. Results show that the pseudo-supervised model has similar or better performance than the reference model trained with ground-truth depth. We also conduct experiments comparing the performance of supervised baseline trained with pseudo depth labels and ground-truth depth labels, which proves that a pseudo-supervised baseline is not a weak choice. The results can be found in Appendix F.

## B TRAINING CONFIGURATIONS

We train our model with one GPU (Nvidia RTX A6000) that has a memory of 48G and the CPU is Intel Xeon Silver 4214R. For each model, doing adversarial training from scratch takes around 70 hours. It includes 20 epochs of training on the KITTI dataset. The fine-tuning of 3 epochs takes about 10 hours. The input resolution of our MDE model is 1024*320 and the original monodepth2 and depthhints models we used for fine-tuning are the official versions trained with both stereo images and videos. In our hardening, we use stereo images with fixed camera pose transformation $T_{t \to s}$. In perturbation generation, we use 10 steps and a step size of $2.5 \cdot \epsilon / 10$ in $L_2$ and $L_\infty$-bounded attacks to ensure that we can reach the boundary of the $\epsilon$-ball from any starting point within it (Madry et al., 2018) and a batch size of 12. In MDE training, the batch size is 32, and the learning rate is 1e-5. We use Adam as the optimizer and other training setups are the same as the original model.

As for the selection of 2D images of objects, as shown in Figure 2 (a) and Figure 2 (b), we have assumptions about the initial relative positions between the target object and the camera (i.e., the 3D coordinates of the center of the object is $(0, 0, z_c)$ in the camera's coordinate system and the viewing angle $\alpha$ of the camera is 0 degree). Hence, for a more realistic and high-quality synthesis, the camera should look at the center of the target object at the same height while taking the 2D image of the object. The width $w$ and height $h$ of the 2D image of the object should be proportional to the physical size $W$ and $H$ of it: $w/W = h/H$. Moreover, when we prepared the 2D image of the object, we also prepared a corresponding mask to "cut out" the main body of the object for projection and we take the object together with its shadow in the mask to preserve reality. Surrounding pixels not covered by the mask are not projected in view synthesis.

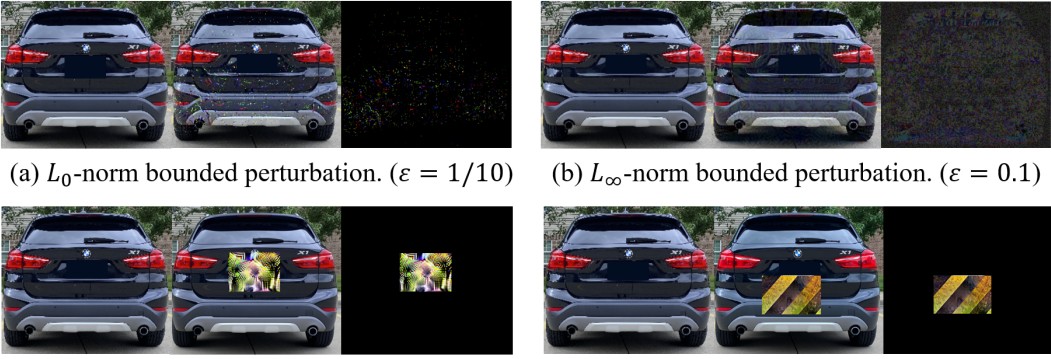

(a) $L_0$-norm bounded perturbation. ($\varepsilon = 1/10$)  (b) $L_\infty$-norm bounded perturbation. ($\varepsilon = 0.1$)

(c) Unbounded adversarial patch attack.  (d) Optimal physical-world patch attack.

Figure 8: Examples of adversarial attacks in our robustness evaluation.

## C  EVALUATION METRICS

The evaluation metrics we used in our evaluation are defined as follows, where we use $X = \{x_1, x_2, ..., x_n\}$ to denote the estimated depth map and $Y = \{y_1, y_2, ..., y_n\}$ to denote the reference depth map and $I()$ is the indicator function that evaluates to 1 only when the condition is satisfied and 0 otherwise.

$$ABSE = \frac{1}{n}\sum_{i=1}^{n}|x_i - y_i| \tag{11}$$

$$RMSE = \sqrt{\frac{1}{n}\sum_{i=1}^{n}(x_i - y_i)^2} \tag{12}$$

$$ABSR = \frac{1}{n}\sum_{i=1}^{n}\left(\frac{|y_i - x_i|}{y_i}\right) \tag{13}$$

$$SQR = \frac{1}{n}\sum_{i=1}^{n}\frac{(y_i - x_i)^2}{y_i} \tag{14}$$

$$\delta = \frac{1}{n}\sum_{i=1}^{n}I(\max\{\frac{x_i}{y_i}, \frac{y_i}{x_i}\} < 1.25) \tag{15}$$

The mean absolute error (ABSE) and root mean square error (RMSE) are common metrics and are easy to understand. Intuitively, the relative absolute error (ABSR) is the mean ratio between the error and the ground truth value, and the relative square error (SQR) is the mean ratio between the square of error and the ground truth value. $\delta$ denotes the percentage of pixels of which the ratio between the estimated depth and ground-truth depth is smaller than 1.25.

## D  ADVERSARIAL ATTACK EXAMPLES

Figure 8 gives examples of the three kinds of adversarial attacks we conducted in our robustness evaluation. The first column is the original object; the second column is the adversarial one and

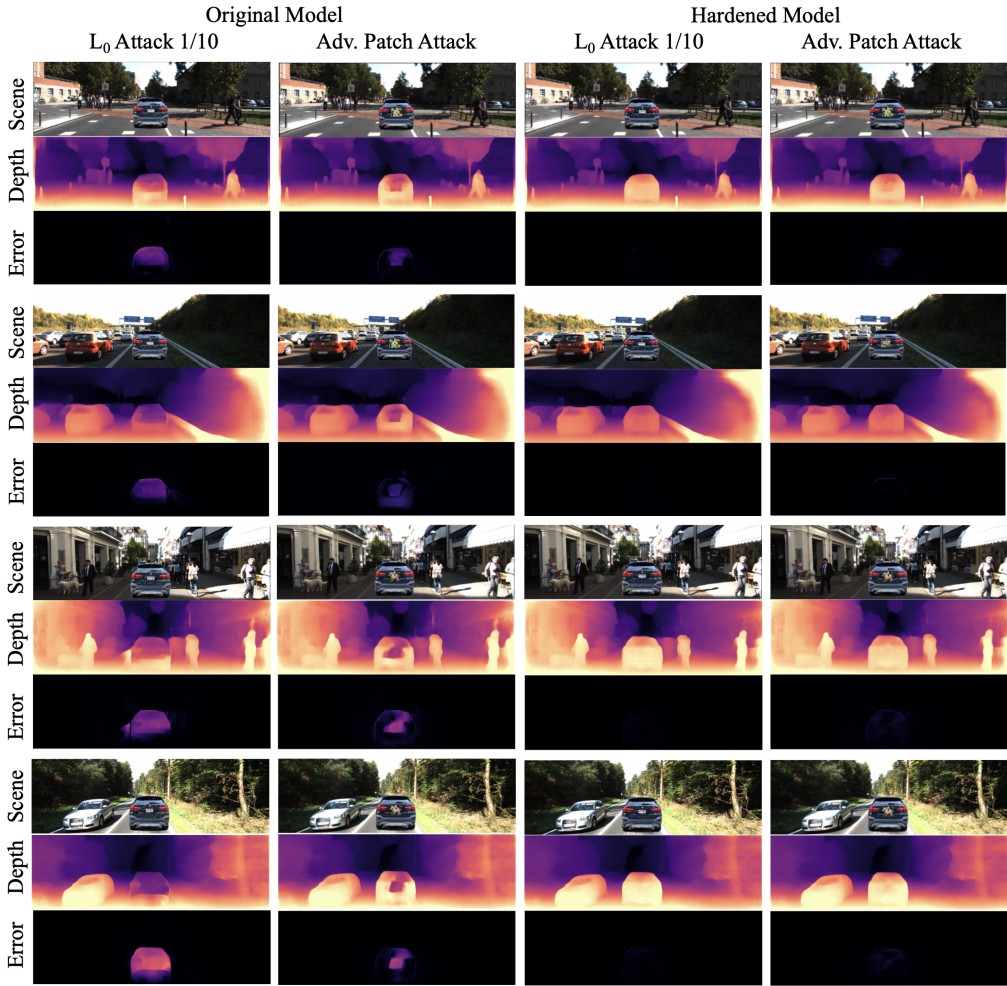

Figure 9: **Qualitative results** of the defensive performance of our hardened model.

the third column is the adversarial perturbations. We scale the adversarial perturbations of the $L_\infty$-norm-bounded attack for better visualization. $L_0$-norm restricts the number of perturbed pixels. $L_\infty$-norm restricts the magnitude of perturbation of each pixel. Adversarial patch attack perturbs pixels within the patch area.

Figure 9 shows the qualitative results of our evaluation. The vertical axis denotes different street views and the horizontal axis denotes different models and attacks. Here we compare the performance of the original Monodepth2 model with our model hardened by **L0+SelfSup**. For each street view, model, and attack, the first row is the adversarial scene (i.e., the scene with the adversarial object), the second row is the corresponding depth estimation and the third row is the depth estimation error caused by the adversarial perturbations. As shown, our method mitigates the effectiveness of attacks significantly and the attacks can hardly cause any adversarial depth estimation error on our hardened model. In addition, our method works on different scenes including complex ones that have multiple pedestrians or vehicles at different speeds. It is because we do not have assumptions about the scene geometry or surrounding objects in our method. Both the scene synthesis and depth estimation are single-image-based and the scene geometry or moving speed will not affect the quality of synthesis.

Table 4: Defence performance of attacks on **various target objects**.

| Models
Objects | Original | | L0+SelfSup
(Ours) | | L0+Sup | | L0+Contras | |
|---|---|---|---|---|---|---|---|---|
| | ABSE↓ | δ↑ | ABSE↓ | δ↑ | ABSE↓ | δ↑ | ABSE↓ | δ↑ |
| BMW SUV Black | 6.079 | 0.512 | **0.248** | **0.988** | 0.949 | 0.934 | 1.642 | 0.552 |
| Toyota Sedan Blue | 4.259 | 0.456 | **0.651** | **0.905** | 1.097 | 0.833 | 1.701 | 0.606 |
| Subaru Sedan White | 4.262 | 0.437 | **1.408** | **0.776** | 2.602 | 0.632 | 2.324 | 0.505 |
| Volvo SUV Grey | 6.379 | 0.535 | **1.565** | **0.889** | 1.859 | 0.836 | 2.476 | 0.552 |
| Traffic Barrier | 4.479 | 0.375 | 1.619 | 0.54 | 2.3 | 0.483 | **0.813** | **0.767** |

*Bold indicates the best performance in each row and underlining means the second best.

Table 5: Performance of the supervised baseline trained with pseudo depth label (`L0+Sup(Pseudo)`) and ground-truth depth label (`L0+Sup(GT)`).

| Attacks | Original | | L0+SelfSup (Ours) | | L0+Sup(Pseudo) | | L0+Sup(GT) | |
|---|---|---|---|---|---|---|---|---|
| | ABSE↓ | δ↑ | ABSE↓ | δ↑ | ABSE↓ | δ↑ | ABSE↓ | δ↑ |
| L0 1/10 | 6.08 | 0.51 | 0.25 | 0.98 | 0.94 | 0.93 | 0.75 | 0.92 |
| L0 1/5 | 8.83 | 0.39 | 0.34 | 0.9 | 1.59 | 0.85 | 0.98 | 0.83 |
| PGD 0.1 | 11.68 | 0.34 | 1.53 | 0.85 | 2.53 | 0.71 | 2.44 | 0.68 |
| PGD 0.2 | 17.1 | 0.23 | 3.46 | 0.69 | 6.14 | 0.50 | 5.11 | 0.32 |

# E    TRANSFER TO OTHER TARGET OBJECTS

The target object under attack may not be used in training. In this experiment, we evaluate how the adversarially trained models perform on unseen target objects. We evaluate with four vehicles and one traffic barrier using the original and the three hardened Monodepth2 models and we perform $L_0$-norm-bounded attack on each object with $\epsilon = 1/10$. Figure 7 shows the examples of target objects that we used in our transferability evaluation. Figure 7(a) is the object we used in adversarial training and others are target objects under attack in the robustness evaluation. Table 4 shows the result. The first column is the target objects under attack and the following columns are the performance of different models. The first object (BMW SUV Black) is an object used in training and our hardened models are most robust on it. The others are unseen objects during training and the hardened models can still mitigate the attack effect a lot compared with the original model, which validates the transferability of our hardened models. `L0+SelfSup` still has the best performance in general.

# F    SUPERVISED BASELINE WITH GROUND TRUTH DEPTH

Although in the scope of this paper, we discuss self-supervised scenarios and assume the ground-truth depth is not available in training, we still conduct experiments comparing the defensive performance of our supervised baseline trained with pseudo-ground truth and that trained with ground-truth depth. We use Monodepth2 as the subject model. The results in Table 5 show that using ground-truth depth (`L0+Sup(GT)`) has a similar performance to using pseudo ground truth (`L0+Sup(Pseudo)`), and they are still worse than our self-supervised approach. Hence, whether to use ground-truth depth is not the bottleneck, and our pseudo-supervised baseline is not a weak choice, which also has significant defensive performance against adversarial attacks.

# G    BROADER IMPACT AND LIMITATIONS

Our adversarial training method hardens the widely used self-supervised monocular depth estimation networks and makes applications like autonomous driving more secure with regard to adversarial attacks. Compared to original training, the hardening of such models with our method does not require additional data or resources and the computational cost is affordable. The adversarial attacks we studied are from existing works and we do not pose any additional threats. Some limitations we could think of about our method are as follows. In our synthesis of different views,

we assume the physical-world object is a 2D image board instead of a 3D model to avoid using an expensive scene rendering engine or high-fidelity simulator and to improve efficiency, which could induce small errors in synthesis though. However, since most physical-world attacks are based on adversarial patches attached to a flat surface, this 2D board assumption is realistic and practical. Precise synthesis should consider lighting factors such as glare, reflections, shadows, etc., but how to do that at a low-cost is an open problem and we leave it as our future work. In addition, although our modeling of the relative positions and viewing angles of the camera and physical object does not cover all real-world conditions, it considers the most common situations. There might be some corner cases in which the adversarial attack could escape from our defense, but we still mitigate the threats in the most common situations and improve the overall security a lot.

