# OpenReview forum: "Adversarial Training of Self-supervised Monocular Depth Estimation against Physical-World Attacks"
_ICLR.cc/2023/Conference — ICLR 2023 notable top 25%_

### Official Review · Reviewer_fV94 · 2022-10-19

**Confidence:** 4
**Correctness:** 4
**Technical Novelty And Significance:** 4
**Empirical Novelty And Significance:** 3
**Recommendation:** 8

**Clarity, Quality, Novelty And Reproducibility:**

The presented method appears to be new and is well written.

In particular, I like the effort that is put into Section 1 to first explain how a 3D attack can be executed and then, to hint that this effort can be reduced by simulating this during training. Therefore, it becomes very clear what kind of attack the paper is focused on, before the derivation of the 3D geometry-based simulation is described.

I can imagine that this paper is a very nice read for everyone who wants to get familiarized with the topic without having too much prior knowledge.

**Strength And Weaknesses:**

Strengths:
+ The proposed method is explained in detail and every derivation is easy to follow
+ The challenges of creating adversarial examples for the presented regression problem is well motivated and convincingly executed
+ The paper use SOTA models as well as SOTA datasets for the training and SOTA attacks for the evaluation

Weaknesses:
- MDE can also be attacked by having an arbitrary painting on a board. This is neither addressed by the approach nor evaluated in the experiments


**Summary Of The Paper:**

The paper addresses the problem of robustness for self-supervised MDE (monocular depth estimation). The focus is on physical world attacks that place an image on a board in front of the camera in order to create an incorrect depth estimation.

The proposed approach follows the adversarial training setting, by including these "board images" during the training phase. Self-supervised MDEs work with stereo cameras where, e.g., the right camera provides the labeling for the left camera. Therefore, a "board image" perturbation on the left image creates a specific perturbation on the right image. The proposed method models this effect from data (left image) to label (right image) via the known relative 3D locations of the cameras.

To create photo-realistic "board images", the approach starts with a realistic 2D image and perturbs it with respect to small 2D changes.
While the goal is to provide small L0 perturbations, this is done in an approximated manner (using tanh as an approximating function for L0)

The experiments start with state-of-the-art MDE models and evaluate the proposed methods on recent MDE attacks.

**Summary Of The Review:**

Overall, I like the ideas, the evaluation and the presentation of the paper very much.
Due to the mentioned weakness, I cannot give the paper the highest rate, but I would like to see this paper presented at ICLR'23.

---

> ### Author Response · Authors · 2022-11-18
> **Official Response to Reviewer fV94**
>
> We appreciate the reviewer's time and constructive comments. We address the primary concerns below.
>
> > W1: MDE can also be attacked by having an arbitrary painting on a board. This is neither addressed by the approach nor evaluated in the experiments.
>
> This is a good perspective and thanks for pointing it out. Due to the fact that the arbitrary paintings are not guided by the gradients from back propagation, it is actually not an effective method of attack. We conduct additional experiments using arbitrary paintings on a patch at the center of the target object, in which we use 500 examples of a single color and 500 examples of random pixels in the patch area. The results indicate that the maximum ABSE caused by the painting among the 1000 examples is 1.142 m on the original model, 0.826 m on our L0+SelfSup model, 1.150 m on the L0+Sup model, and 1.025 m on the L0+Contras model. The results demonstrate the attacking effectiveness of arbitrary paintings, but the performance is limited. Our hardened models can further reduce the effect.

---

> > ### Comment · Reviewer_fV94 · 2022-11-23
> > **Reponse to the Rebuttal**
> >
> > Thanks a lot to the authors for reporting the additional experiment.
> >
> > It did not change my accept vote in any sense.

---

### Official Review · Reviewer_9eWp · 2022-10-23

**Confidence:** 4
**Correctness:** 3
**Technical Novelty And Significance:** 2
**Empirical Novelty And Significance:** 2
**Recommendation:** 6

**Clarity, Quality, Novelty And Reproducibility:**

I think this paper proposes a new strategy for the adversarial training of MDE networks. But the experiments are not sufficient to support the effectiveness of the proposed strategy.
And this paper has not provided enough materials to guarantee the reproducibility.

**Details Of Ethics Concerns:**

None.

**Strength And Weaknesses:**

Strength:

This paper propose a self-supervised adversarial training strategy for MDE with view synthesis, and the results is obviously better than the supervised learning baseline and a contrastive learning baseline.

Weakness:

1. The proposed approach is limited on the MDE task with self-supervised setting, while practical depth estimation network are trained with supervised learning. Can the proposed approach be utilized for the situation where the ground truth is provided?

2. The experiments are only conducted on the KITTI which is an outdoor dataset, and I wonder if the robustness can be hold for the indoor scenes, e.g., conduct the experiments on the NYUD dataset.

3. Monodepth2 and DepthHints are not SOTA MDE frameworks. Although the authors have proved the superiority of the proposed self-supervised adversarial training with these two networks compared with supervised loss, more experiments should be conducted with SOTA MDE networks, like [A, B]

[A]The Temporal Opportunist: Self-Supervised Multi-Frame Monocular Depth, CVPR2021
[B]Exploiting Pseudo Labels in a Self-Supervised Learning Framework for Improved Monocular Depth Estimation, CVPR2022

4. The meaning of each method setting should be well shown in Tables 1 and 2.

5. This is one important baseline to compare: minimize the outputs of trained MDE networks with the inputs of clean and adversarial samples.

**Summary Of The Paper:**

This paper proposes a new adversarial training method for self-supervised MDE models without depth ground truth. The training utilizes the the reconstruction consistency from one view to the other view. And to improve the adversarial robustness against physical-world attacks, $L_0$-norm-bounded perturbations are utilized in training. Results on two MDE networks demonstrate the robustness achieved by the proposed methods.

**Summary Of The Review:**

This paper needs more experiments to demonstrate their importances and effectiveness. And the paper should be organized with more details.

---

> ### Author Response · Authors · 2022-11-18
> **Official Response to Reviewer 9eWp (Part 1: W1, W2 & Table R3)**
>
> We appreciate the reviewer's time and constructive comments. We address the primary concerns below.
>
> > W1: The proposed approach is limited on the MDE task with self-supervised setting, while practical depth estimation network are trained with supervised learning. Can the proposed approach be utilized for the situation where the ground truth is provided?
>
> Thanks for the insightful question. Please allow us to elaborate on this issue. Firstly, the self-supervised MDE has been widely adopted by production-grade autonomous driving systems (e.g., Tesla Autopilot [1]) in the industry because it does not require depth labels and can be trained with pre-existing videos at a low cost, so the robustness of such models is of critical importance nowadays. Secondly, our self-supervised adversarial training method is applicable to **ALL** MDE models regardless of their original training approach because we do not have assumptions about the MDE network structure, and basically, we use the MDE model to provide per-pixel depth estimation and update the model parameters to minimize the photo reconstruction loss and encourage the MDE model to make accurate depth estimations for adversarial examples. For MDE models trained in a supervised manner, people can also harden these models with the proposed self-supervised approach if the original depth labels are no longer available (e.g., customers who do not have access to the original training set of the model they bought.). Thirdly, if the ground truth is provided, the model can be hardened with the conventional supervised adversarial training approach and follow the original training scheme specifically designed for each model. Although our self-supervised training framework may not help in such cases, the proposed physical-world-oriented view synthesis techniques using various object and camera settings and the $L_0$-bounded adversarial perturbation generation can also be applied to improve the adversarial robustness against physical-world attacks. Thanks again for the insightful question, and we will add the discussion to the next version of our paper.
>
>
> > W2: The experiments are only conducted on the KITTI which is an outdoor dataset, and I wonder if the robustness can be hold for the indoor scenes, e.g., conduct the experiments on the NYUD dataset.
>
> Thank you for your suggestion. Although we focus on outdoor scenarios like autonomous driving, a domain where self-supervised MDE is widely adopted (e.g., Tesla Autopilot), our technique hardens the MDE networks and does not have any assumptions about indoor or outdoor scenes. Actually, both Monodepth2 and DepthHints models have been proven to be directly applicable on indoor scenes (e.g., NYU-Depth-v2 Dataset) without any retraining [2-3], which implies the indoor-scene capability of our hardened models. We further conduct additional experiments with indoor scenes in the NYU-Depth-v2 Dataset. We launch adversarial attacks in a square area at the center of each scene to mimic a physical patch in the scene and evaluate the defensive performance of the original and the hardened monodepth2 models. Results are shown in Table R3 below. As shown, our hardened models reduce the depth estimation error caused by the attack significantly, and the model trained with our self-supervised method has the best defensive performance. Hence, the robustness of our hardened model still holds for indoor scenes. We incorporate the results in Appendix L of our revised paper.
>
>
> Table R3: Defensive performance on the indoor scenes with the NYU-Depth-v2 Dataset.
> | **Attacks** | **Original** |  | **L0+SelfSup (Ours)** |  | **L0+Sup** |  | **L0+Contras** |  |
> |:---:|:---:|:---:|:---:|:---:|:---:|:---:|:---:|:---:|
> |  | **ABSE** | **$\delta\uparrow$** | **ABSE** | **$\delta\uparrow$** | **ABSE** | **$\delta\uparrow$** | **ABSE** | **$\delta\uparrow$** |
> | L0 1/20 | 1.243 | 0.772 | **0.099** | **0.993** | 0.363 | 0.976 | 0.695 | 0.812 |
> | L0 1/10 | 3.866 | 0.54 | **0.164** | **0.985** | 0.49 | 0.924 | 1.12 | 0.733 |
> | PGD 0.05 | 1.784 | 0.727 | **0.12** | **0.986** | 0.456 | 0.849 | 1.641 | 0.717 |
> | PGD 0.1 | 4.598 | 0.426 | **0.288** | **0.912** | 0.962 | 0.775 | 4.779 | 0.42 |

---

> ### Author Response · Authors · 2022-11-18
> **Official Response to Reviewer 9eWp (Part 2: W3-6 & Table R4)**
>
> > W3: Monodepth2 and DepthHints are not SOTA MDE frameworks. Although the authors have proved the superiority of the proposed self-supervised adversarial training with these two networks compared with supervised loss, more experiments should be conducted with SOTA MDE networks like [A, B].
>
> Thank you for the suggestion. Monodepth2 and DepthHints are the two popular and representative self-supervised MDE networks, and they are widely used as baselines in the literature, so we use them as our subject networks. However, we do not have any assumptions about the MDE network structure, and our method should also work on SOTA MDE models. During rebuttal, we tried to experiment with [A, B]. We found that [B] does not open-source their pre-trained models while our technique fine-tunes original models. We hence focused on experimenting with Manydepth [A]. We used our $L_0$-bounded perturbation with self-supervised adversarial training to harden the original Manydepth model, and Table R4 shows the defensive performance of the original and hardened models. As shown, the original models are still vulnerable to both $L_0$-bounded and PGD attacks. The model hardened with our techniques is a lot more robust. The mean depth estimation error is reduced by over 80%, which validates that our techniques are generic and also work on SOTA MDE models. We incorporate the results in Appendix L of our revised paper.
>
> Table R4: Defensive performance of the original and hardened ManyDepth models.
> | **Attacks** | **Original** |  | **L0+SelfSup** |  |
> |:---:|:---:|:---:|:---:|:---:|
> |  | **ABSE** | **$\delta\uparrow$** | **ABSE** | **$\delta\uparrow$** |
> | L0 1/20 | 1.554 | 0.78 | **0.221** | **0.996** |
> | L0 1/10 | 2.684 | 0.662 | **0.301** | **0.994** |
> | PGD 0.05 | 7.112 | 0.417 | **1.270** | **0.855** |
> | PGD 0.1 | 9.452 | 0.339 | **1.614** | **0.83** |
>
> > W4: The meaning of each method setting should be well shown in Tables 1 and 2.
>
> Thanks for the suggestion, and we have added explanations to the footnotes of Table 1 and Table 2 in our updated paper.
>
> > W5: This is one important baseline to compare: minimize the outputs of trained MDE networks with the inputs of clean and adversarial samples.
>
> Thanks for your comments. We clarify that we included it as the supervised baseline in our evaluation (cf. Table1 and Table2). In the supervised baseline, the outputs of the MDE model with the inputs of clean samples were used as pseudo-depth labels whereas the ground-truth depth was not available in the self-supervised scenario. Then we supervised the model training with the pseudo-labels and updated model parameters to minimize the difference between pseudo-labels and the model output of corresponding adversarial samples. Our results showed that the models hardened with our method are more robust against various attacks and have better benign performance than the baseline.  Details of our baselines can be found in Appendix A and Figure 6 of our paper.
>
> > W6:  The experiments are not sufficient to support the effectiveness of the proposed strategy. This paper has not provided enough materials to guarantee the reproducibility.
>
> Thanks for the constructive comment. During the rebuttal phase, we added more experiments based on your suggestions (cf. W2-4) to validate the efficacy of the proposed method. In addition, we want to respectfully reiterate that we made a reproducibility statement in Section 6 of our paper, which describes our efforts to make the work reproducible. It includes detailed experimental setups in both the main text and appendix, the source code, and the video of our physical-world experiments in the supplementary materials. If there are any specific things unclear for reproduction, please let us know, and we will be happy to address them in our paper.
>
>
> [1] Andrej Karpathy. https://youtu.be/hx7BXih7zx8?t=1334 .
>
> [2] Peng, Rui, et al. "Excavating the potential capacity of self-supervised monocular depth estimation." CVPR. 2021.
>
> [3] Zhou, Kaichen, et al. "DevNet: Self-supervised Monocular Depth Learning via Density Volume Construction." ECCV. 2022.

---

### Official Review · Reviewer_jvhn · 2022-10-24

**Confidence:** 2
**Clarity, Quality, Novelty And Reproducibility:** The novelty is somewhat limited. Howe…
**Correctness:** 3
**Technical Novelty And Significance:** 2
**Empirical Novelty And Significance:** 2
**Recommendation:** 6

**Strength And Weaknesses:**

A working pipeline for the MDE task under the self-supervised training.
The image synthesize contains some issue.

**Summary Of The Paper:**

This paper targets at the task of MDE against the physical attack in a self-supervised manner. This paper proposes a reconstruction pipeline that pasting the projection of a 3D object onto two views and perturbate one of them, and then minimize the reconstruction error for the adversarial training. In this way, the ground-truth labels are no longer required. A loss is proposed to mimic physical attack.
Experiments are conducted on the KITTI with several attack mode, white box, black box and physical-world attack.

**Summary Of The Review:**

Combine figure 1 and figure 2, figure 2 can be a subfigure of figure 1.

When summarize the contribution, pls make them concise. Some part of the description of summarized contribution can be moved out to the introduction part.

“the image in one view can be transformed to yield the image from the other view. For instance, I_t can be acquired by shifting I_s to the left”. This is not that intuitive, which requires dense depth, and the accurate image inpainting process of the possible `holes' caused by scene disparities during the view shift. The situation becomes more challenging when using consecutive frames of video.

There are image issues, such as inconsistent environment lighting between the scene and object, if we pasting the projection of a 3D object directly onto the image plane. Using a virtual rendering engine, as mentioned in the paper, faces the similar issue.

Using the warp figures is not that informative for illustrate a figure.

---

> ### Author Response · Authors · 2022-11-18
> **Official Response to Reviewer jvhn**
>
> We appreciate the reviewer's time and constructive comments. We address the primary concerns below.
>
> > W1:Combine figure 1 and figure 2, figure 2 can be a subfigure of figure 1.
>
> Thanks for the suggestion. We have combined the two figures in our revised paper.
>
> > W2: When summarize the contribution, pls make them concise. Some part of the description of summarized contribution can be moved out to the introduction part.
>
> We appreciate your suggestion. To maintain a concise summary of the contribution, we have modified the contribution section of the introduction in our revised paper and moved some content to the introduction section.
>
> > W3: “the image in one view can be transformed to yield the image from the other view. For instance, I_t can be acquired by shifting I_s to the left”. This is not that intuitive, which requires dense depth, and the accurate image inpainting process of the possible `holes' caused by scene disparities during the view shift. The situation becomes more challenging when using consecutive frames of video.
>
> Thanks for your comment. Please allow us to elaborate on this issue. Yes, we do use the dense depth (i.e., the per-pixel depth estimation output of the MDE model) and camera pose transformation (e.g., a fixed matrix in the case of stereo images or the output of the camera transposing model $TP$ with consecutive video frames as input) to reconstruct one view from the other. Section 3.3 describes the process in detail. Due to the fact that two views are shot at two close-by camera positions, either with stereo cameras or two consecutive video frames, the “holes” caused by view shifting are very sparse, and we use interpolation to fill any potential “holes”. We have revised the text based on your comment regarding the conceptual nature of the introduction's description: “Intuitively, $I_t$ can be acquired by shifting $I_s$ to the left.”
>
> > W4: There are image issues, such as inconsistent environment lighting between the scene and object, if we pasting the projection of a 3D object directly onto the image plane. Using a virtual rendering engine, as mentioned in the paper, faces the similar issue.
>
> This is a good point, and we appreciate the reviewer for pointing that out. Realistic synthesis is a complex problem, and there are a lot of factors to consider such as lightning, reflection, shadows, etc. 3D rendering engines could approximate some factors, but with a high cost. We have considered the lighting in a simplified way in our view synthesis: adjust the brightness of the 2D object image according to the average brightness of the background image before pasting. Although it is not perfect, our method is lightweight and serves as a solid foundation for physical-world-oriented adversarial training and contributes to the improvement of model robustness. More precise synthesis considering glare, shadows, and reflection at a low cost will be our future endeavor, and we have added this point to the limitations of our paper.
>
> > W5: Using the warp figures is not that informative for illustrate a figure.
>
> Thanks for the suggestion. We have modified our format to avoid using wrap figures and provide the illustration more clear and informative.

---

> ### Author Response · Authors · 2022-11-22
> **Message from authors**
>
> Since we are still in the second phase of the discussion, we would like to further discuss with the reviewer. Has our revision helped to resolve your concerns and helped you reevaluate our contribution? If there are any new concerns, we will be happy to address them.
>
> We are looking forward to hearing from you.
>
> Thanks.

---

### Official Review · Reviewer_QLeq · 2022-10-28

**Confidence:** 4
**Correctness:** 3
**Technical Novelty And Significance:** 3
**Empirical Novelty And Significance:** 4
**Recommendation:** 8

**Clarity, Quality, Novelty And Reproducibility:**

The structure of this paper is well-organized and clear. But there are still some parts that could be more specific and precise. This paper is likely to provide new methods to improve adversarial robustness against physical-world attacks. However, more experimental evaluation could be added to make their technique more convincing.

**Strength And Weaknesses:**

Strength:
1. The idea is interesting and seems novel. The structure of this paper is clear.
2. The paper appears to be technically sound and achieves some improvements over the existing methods.

Weakness:
1. The support of not using L_2 L_inf for adversarial training should be included in the experiment sections.
2. The experiment is inadequate. This paper only uses L0-norm-bounded attacks and PGD attacks for black-box and white-box attacks. More types of attacks should be added to demonstrate the robustness of the model.  Existing experiments only contain the PGD attacks while ignoring more powerful attacks like Autoattack, which affects the convincing. Besides, are the adversarially trained models still robust to the natural perturbation-based adversarial attacks like adversarial blur attacks, and adversarial light attacks?
3. The reason for the baseline selection is unclear. The authors can further explain why they were chosen as baselines and at what point they contrast with the newly proposed model.
4. It claims the synthesized views are realistic. However, there are no experimental supports.
5. How about the influences of inaccurate projections on view synthesis?
6. The pictures in the article are somewhat misleading. The paper mentions that Figure 1 provides a conceptual illustration of their technique. It seems that in Figure 1 Cs and Ct are provided by two cameras, but the technique uses view synthesis in fact. Although they mention it in section 3, it would avoid misunderstanding if the authors could stress this in the figure.

**Summary Of The Paper:**

This work proposes an adversarial training method for self-supervised MDE models based on view synthesis without using ground-truth depth. It uses L0-norm-bounded perturbation in training to improve adversarial robustness against physical-world attacks. The experiments demonstrate the proposed methods can maintain similar performance on the raw data while defending PGD Patch attacks effectively. The idea of using self-supervised MDE is interesting and the novel framework seems novel. My main concerns are about the detailed designs and experiments.

**Summary Of The Review:**

This work proposes an adversarial training method for self-supervised MDE models based on view synthesis without using ground-truth depth. It uses L0- norm-bounded perturbation in training to improve adversarial robustness against physical-world attacks. The paper is well organized but the presentation has some details that could be improved. More experiments should be added to justify their technique.

---

> ### Author Response · Authors · 2022-11-18
> **Official Response to Reviewer QLeq (Part 1: W1 & Table R1)**
>
> We appreciate the reviewer's time and constructive comments. We address the primary concerns below.
>
> > W1: The support of not using L_2 L_inf for adversarial training should be included in the experiment sections.
>
> Thanks for the comment. We did use the $L_\infty$-bounded attack for adversarial training in our experiments, which is the original PGD attack [1] as stated in the training setup part of the experimental setup section in our paper. We did not choose $L_2$-bounded attacks because [1] has demonstrated that models hardened with $L_\infty$-bounded perturbations are also robust against $L_2$-bounded attacks. In addition, physical-world attacks with adversarial patches have more resemblance to $L_0$-bounded attacks that only restrict the ratio of perturbed pixels rather than the magnitude of the perturbation. To address your concerns more thoroughly, we conduct additional experiments to evaluate the robustness of our models against $L_2$-bounded attacks. Results are shown in the first three rows of Table R1, our models hardened with $L_0$-bounded perturbation are still robust to $L_2$-bounded attacks with ABSE less than 1 meter, and our method still outperforms the baselines. The clarification can be found in Appendix B of our revised paper.
>
> Table R1: Additional Attacks to our hardened models.
> | **Attacks** | **Original** |  | **L0+SelfSup (Ours)** |  | **L0+Sup** |  | **L0+Contras** |  |
> |:---|:---:|:---:|:---:|:---:|:---:|:---:|:---:|:---:|
> |  | **ABSE** | **$\delta\uparrow$** | **ABSE** | **$\delta\uparrow$** | **ABSE** | **$\delta\uparrow$** | **ABSE** | **$\delta\uparrow$** |
> | L2-PGD $\epsilon$=8 | 1.403 | 0.76 | **0.294** | **0.996** | 1.161 | 0.741 | 0.66 | 0.919 |
> | L2-PGD $\epsilon$=16 | 6.491 | 0.522 | **0.597** | **0.984** | 2.516 | 0.479 | 1.437 | 0.734 |
> | L2-PGD $\epsilon$=24 | 13.018 | 0.354 | **0.932** | **0.913** | 3.613 | 0.387 | 2.92 | 0.7 |
> | APGD $\epsilon$=0.05 | 5.557 | 0.739 | **0.423** | **0.976** | 1.614 | 0.793 | 2.71 | 0.859 |
> | APGD $\epsilon$=0.1 | 10.216 | 0.46 | **0.928** | **0.945** | 3.279 | 0.603 | 4.578 | 0.793 |
> | Square $\epsilon$=0.1 N=5000 | 0.924 | 0.924 | **0.422** | **0.991** | 0.712 | 0.934 | 0.568 | 0.973 |
> | Gaussian Blur | 0.323 | 0.996 | **0.191** | **0.997** | 0.288 | 0.997 | 0.264 | 0.997 |
> | AdvLight | 0.512 | 0.988 | **0.493** | **0.991** | 0.513 |0.987  |0.504  | 0.988 |

---

> ### Author Response · Authors · 2022-11-18
> **Official Response to Reviewer QLeq (Part 2: W2 & W3)**
>
> > W2: The experiment is inadequate. This paper only uses L0-norm-bounded attacks and PGD attacks for black-box and white-box attacks. More types of attacks should be added to demonstrate the robustness of the model. Existing experiments only contain the PGD attacks while ignoring more powerful attacks like Autoattack, which affects the convincing. Besides, are the adversarially trained models still robust to the natural perturbation-based adversarial attacks like adversarial blur attacks, and adversarial light attacks?
>
> We have substantially improved the experiments by adding eight more attacks based on the constructive comment. But we want to first clarify that, in addition to the $L_0$-bounded attacks and PGD attacks, we had also evaluated our method with a physical-world-oriented adversarial patch attack [2] and a SOTA physical-world attack against MDE with an optimal adversarial patch [3] (as stated in the Attacks part of our Evaluation setup section). Examples of these attacks can be found in Figure 8 in the Appendix and the video of physical-world experiments that provided illustrations in the supplementary material.
>
> In the new experiments, Autoattack [4] is an ensemble of different attacks (APGD, FAB, and Square Attack), and it is used as a benchmark to evaluate the robustness of image classification models. During rebuttal, we conducted additional experiments with these attacks in Autoattack that can be adapted for MDE (i.e., per-pixel-based regression task), such as the APGD attack and the black-box Square attack [5]. Results can be found in the 4th to 6th rows in Table R1. Since they are not designed specifically for regression tasks, they are not necessarily more powerful than PGD. For example, the Square attack with 5000 queries only causes less than 1 m error, even on the original Monodepth2 model. Our model is more robust than the original model in all cases, and our method outperforms all other methods. Attacks with an $L_\infty$-bound of 0.1 can only cause less than 1 m depth estimation error on our hardened models while more than 10 m on the original model. We have updated our paper to include the experiments with more attacks in Appendix J.
>
> The optimal adversarial patch attack [3] uses a style-transfer-based method to camouflage adversarial patterns into natural styles like rusty or dirty, which is a kind of natural perturbation-based adversarial attack mentioned by the reviewer and targets MDE specifically. Our evaluation results in Section 4.2-Physical-world Attack had shown the effectiveness of our method against the attack. We also conducted additional experiments to test our techniques against the blur [6] and light [7] attacks mentioned by the reviewer. Results are reported in the 7th and 8th rows in Table R1. As shown, our models are robust against them, and these attacks are not as effective as those gradient-based attacks like PGD, APGD, optimal patch attack [3], etc., on the task of MDE.
>
> > W3: The reason for the baseline selection is unclear. The authors can further explain why they were chosen as baselines and at what point they contrast with the newly proposed model.
>
> Thanks for the comment. As we stated in the baseline part of our experimental setup section, there are no direct baselines available since we are the first to focus on self-supervised adversarial training of MDE without using the ground-truth depth, so we adapted a SOTA contrastive learning-based adversarial training and a traditional supervised training-based adversarial training to MDE. The contrastive learning baseline is SOTA in the domain of self-supervised adversarial training, and the supervised baseline represents applying the conventional adversarial training method to the self-supervised domain with pseudo-ground truth. Details of our baselines can be found in Appendix A of our paper.

---

> ### Author Response · Authors · 2022-11-18
> **Official Response to Reviewer QLeq (Part 3: W4, W5 & Table R2)**
>
> > W4: It claims the synthesized views are realistic. However, there are no experimental supports.
>
> Thanks for the valuable suggestion. We have added more results and a human study based on the comment to validate our design. There were some visualized examples of our view synthesis in the paper, such as $I_t$ and $I_s$ in Figure 1b and the images in Figure 9. To further demonstrate the quality of our view synthesis, we show more examples of the synthesized images in Figure 11 of our revised paper. The target object (i.e., the black SUV) is synthesized into different views with various distances. To evaluate the veracity of these images, we use Amazon Mechanical Turk to conduct human evaluations of our synthesized images. Participants are requested to evaluate the quality of the synthesized objects from 4 distinct perspectives: size, consistency of location, lighting, and overall quality. The score ranges from 1 to 10 for each metric. We use the score of 1 to indicate scenes synthesized with random projections and the score of 10  for real scenes taken by stereo cameras. Examples of perfect (score of 10) and unrealistic  (score of 1) scenes for each metric are provided as references (See Figure 12). The results of the experiment with 100 participants can be found in Table R2. In each row, we show the number of participants who gave a score within the corresponding range and summarize the average in the last row. As demonstrated, We got an average score of 7.7 regarding the size, 7.21 regarding the location,  7.43 regarding the lighting, and 7.74 regarding the overall quality. Compared with the real scene, our synthesized scene is slightly inferior but still much better than random projection. More importantly, it works well to harden the model against physical-world attacks at a low cost.
>
> Table R2: Human evaluations of the quality of our synthesized images.
> | **Score Range** | **Size** | **Location** | **Lightning** | **Overall** |
> |:---:|:---:|:---:|:---:|:---:|
> | **1-2** | 0 | 0 | 1 | 0 |
> | **3-4** | 4 | 6 | 6 | 3 |
> | **5-6** | 16 | 24 | 23 | 18 |
> | **7-8** | 42 | 48 | 40 | 45 |
> | **9-10** | 38 | 22 | 30 | 34 |
> | **Total** | 100 | 100 | 100 | 100 |
> | **Average Score** | 7.7 | 7.21 | 7.43 | 7.74 |
>
> > W5: How about the influences of inaccurate projections on view synthesis?
>
> Thanks for the comment. During rebuttal, we conducted an additional experiment to evaluate the influence of inaccurate projections. According to our experiments, inaccurate projections on view synthesis would cause blurriness on the edges of the object (see Figure 13) while doing depth estimation. We have added the details of this experiment to Appendix L of our updated paper:
>
> “As stated in Section 3.1, we project a 2D image of the target object to two adjacent views considering the camera pose transformation $T_{t\rightarrow s}$ between $C_s$ and $C_t$ (Equation 3 and 4), then we enable the self-supervised training of MDE network by reconstructing $I_t$ from $I_s$ using the estimated depth $D_{I_t’}$ (Equation 9). In this section, we explore what influence the inaccurate projections of the two camera views would have on training results. In this experiment, we only consider half of the camera pose transformation (i.e., $T_{t\rightarrow s}/2$) while projecting the object to $I_s$ instead of the true value $T_{t\rightarrow s}$, which leads to a wrong $I_s$, intentionally. For example, the distance between the stereo cameras in the KITTI dataset is 0.54 m, but we use 0.27 m to synthesize $I_s$. $I_t$ is still correctly synthesized. We use $L_0$-bounded perturbation with $\epsilon=1/10$ and self-supervised training to harden the Monodepth2 model. Figure 13 shows the result. As shown, the edges of the estimated depth of the target object is blurred when the MDE model is trained with inaccurate projection. It remains clear for the model trained with accurate projection.”

---

> > ### Comment · Reviewer_QLeq · 2022-11-22
> > **Response to the rebuttal**
> >
> > Thanks for the efforts on the rebuttal. I still have two questions about the quality of synthesized views. First, as shown in Figure 1, Figure 3, and Figure 7, there are some background remnants at the four corners of board A. However, these remnants disappear at the synthesized views, i.e, in Figures 11 and 12. How could these be achieved? According to the method in Section 3.1 and Figure 2, the pixels in board A are all projected. Second, the human study is not a convincing way to validate the realism. I understand this is not easy. However, the visualizations are only based on one board. Could you show more results with different boards?

---

> > > ### Author Response · Authors · 2022-11-23
> > > **More explanations and results**
> > >
> > > Thanks a lot for your continuous help in improving our paper. We answer your new questions below.
> > >
> > > > W1: First, as shown in Figure 1, Figure 3, and Figure 7, there are some background remnants at the four corners of board A. However, these remnants disappear at the synthesized views, i.e, in Figures 11 and 12. How could these be achieved? According to the method in Section 3.1 and Figure 2, the pixels in board A are all projected.
> > >
> > > Thanks for the insightful question. In the second paragraph of Appendix B, we stated, “Moreover, when we prepare the object mask (to cut out the object from the 2D image), we take the object together with its shadow. ” That is, when we prepared the 2D image of the object ($A$), we also prepared a corresponding mask to “cut out” the main body of the object for projection. Those remnants surrounding the object are hence not projected onto background scenes.
> > >
> > >  Formally, let $M \\in \\{0, 1 \\}^{h \\times w}$ be the object mask of $A$. In $M$, the area of the object's body is filled with 1, and the remnants are 0. The mask can be generated automatically via off-the-shelf image matting models like Background-Matting [1]. In Equation 3 and Equation 4, we assumed $(u^A, v^A)$ are pixels of the object’s body. In other words, we only project those pixels $(u^A, v^A)$ where $M[u^A, v^A] = 1$ during synthesis. More precisely, there is one more condition in the first row of  Equation 3 and Equation 4:  $M[u^A, v^A] = 1$. Thanks for the question again, and we will clarify it in the next version of our paper.
> > >
> > >
> > > > Second, the human study is not a convincing way to validate the realism. I understand this is not easy. However, the visualizations are only based on one board. Could you show more results with different boards?
> > >
> > > Thanks for the suggestion. We have added more results with different objects in the next version of our paper. Since we are not allowed to revise the paper now, the additional figure can be found here: https://drive.google.com/uc?id=1aaMf5P6bVvw_96GVuQ3wJndrJUsGUcXk.
> > >
> > > [1] Lin, Shanchuan, et al. "Real-time high-resolution background matting." In CVPR. 2021.

---

> > > ### Author Response · Authors · 2022-12-01
> > > **Message from Authors**
> > >
> > > Dear reviewer QLeq,
> > >
> > > We would like to hear if our explanations and additional results address your new questions well. We would appreciate it if we could have further discussions if there is anything unclear.
> > >
> > > Thanks.

---

> > > ### Author Response · Authors · 2022-12-08
> > > **Message from Authors**
> > >
> > > Dear reviewer QLeq,
> > >
> > > Thank you again for your kind review and comments. We believe we have addressed your concerns adequately. Since the second stage of discussion is coming to an end, we therefore respectfully ask you to review our responses once more and determine whether there are any additional concerns. We sincerely hope that we will be able to use the remaining time to engage in an open dialogue with domain experts to enhance the quality of our work.
> > >
> > > Thanks for your valuable time in advance.
> > >
> > > Authors.

---

> > > > ### Comment · Reviewer_QLeq · 2022-12-08
> > > > **Response to rebuttal**
> > > >
> > > > Thanks for your efforts. The main concerns are addressed. I would like to suggest accept it.

---

> ### Author Response · Authors · 2022-11-18
> **Official Response to Reviewer QLeq (Part 4: W6, W7 & References)**
>
> > W6: The pictures in the article are somewhat misleading. The paper mentions that Figure 1 provides a conceptual illustration of their technique. It seems that in Figure 1 Cs and Ct are provided by two cameras, but the technique uses view synthesis in fact. Although they mention it in section 3, it would avoid misunderstanding if the authors could stress this in the figure.
>
> Thanks for the suggestions. We have combined previous Figure 1 and Figure 2 to be one and use a sub-caption to stress the conceptual illustration point.
>
> > W7: Overall comments in “Clarity, Quality, Novelty And Reproducibility” and “Summary Of The Review”
>
> Thanks for the positive comments. We have addressed all the concerns point-to-point in the aforementioned responses.
>
> [1] Aleksander Madry, Aleksandar Makelov, Ludwig Schmidt, Dimitris Tsipras, and Adrian Vladu. Towards deep learning models resistant to adversarial attacks. In ICLR, 2018.
>
> [2] Alwyn Mathew, Aditya Prakash Patra, and Jimson Mathew. Monocular depth estimators: Vulnerabilities and attacks. arXiv preprint arXiv:2005.14302, 2020.
>
> [3] Zhiyuan Cheng, James Liang, Hongjun Choi, Guanhong Tao, Zhiwen Cao, Dongfang Liu, and Xiangyu Zhang. Physical attack on monocular depth estimation with optimal adversarial patches. In ECCV, 2022
>
> [4] Croce, Francesco, and Matthias Hein. "Reliable evaluation of adversarial robustness with an ensemble of diverse parameter-free attacks." In ICML, 2020.
>
> [5] Andriushchenko, Maksym, et al. "Square attack: a query-efficient black-box adversarial attack via random search." In ECCV, 2020.
>
> [6] Rauber, Jonas, et al. "Foolbox native: Fast adversarial attacks to benchmark the robustness of machine learning models in pytorch, tensorflow, and jax." Journal of Open Source Software 5.53 (2020): 2607.
>
> [7] Duan, Ranjie, et al. "Adversarial laser beam: Effective physical-world attack to dnns in a blink." In CVPR. 2021.

---

> ### Author Response · Authors · 2022-11-22
> **Message from Authors**
>
> Since we are still in the second phase of the discussion, we would like to hear the reviewer's opinion about our revision. Has our revision helped to resolve the reviewer's concerns and helped the reviewer reevaluate our contribution? If there are any new concerns, we still have time to discuss and we will be happy to address them.
>
> We are looking forward to hearing from the reviewer.
>
> Thanks.

---

### Author Response · Authors · 2022-11-18
**Paper Revision Summary**

We thank all the reviewers for their insightful questions and constructive suggestions! We are glad that the reviewers found our paper “technically sound”, “well written”,  “structure of this paper is clear”, “proposed method is explained in detail”, “ every derivation is easy to follow”, “well motivated and convincingly executed”, and “a very nice read”.

Below is a summary of paper updates, and we also mark the updates in the paper with blue color:
1. **[Section 1: Introduction]** Combine Figure 1 and Figure 2, and use a subcaption to stress the conceptual illustration of Figure 1a.
2. **[Section 1: Introduction]** Move the description of our contributions on $L_0$-norm-based adversarial training to the main text and keep summaries of contributions more concise.
3. **[Section 4: Evaluation]** Add footnotes to Table 1 and Table 2 to explain the meaning of model names.
4. **[Section 4: Evaluation]** Change the format to avoid using wrap figures for Figure 5.
5. **[Appendix B: Training Configurations]** Add support for not using the $L_2$ norm in adversarial training.
6. **[Appendix D: Adversarial Attack Examples]** Add one more example of the optimal physical-world patch attack in Figure 8.
7. **[Appendix J: Robustness Against more Attacks]** Add experiments on eight more attacks, including three $L_2$-bounded attacks [1], two APGD attacks [2],  a black-box Square attack [3], a Gaussian Blur attack [4] and a Laser Light attack [5]), to evaluate the robustness of our models. Results are reported in Table 9.
8. **[Appendix K: Quality of the View Synthesis]** Add experiments to evaluate the quality of our view synthesis, including visualization of various synthesized images (Figure 11), perfect and baseline samples for references (Figure 12) and a human study using Amazon Mechanical Turk (Table 10).
9. **[Appendix L: Influence of Inaccurate Projections]** Add experiments to explore the influence of using inaccurate projections in adversarial training. Figure 13 shows the result.
10. **[Appendix M: Extension to Indoor Scenes and Advanced Networks]** Add experiments to validate the effectiveness of our method on indoor scenes and more advanced MDE networks. Table 11 and Table 12 report the results.
11. **[Appendix N: Broader Impact and Limitations]** Add discussion about precise synthesis regarding lightning.

Please also let us know if there are other questions, and we really look forward to the discussion with the reviewers to further improve our paper. Thank you.

[1] Aleksander Madry, Aleksandar Makelov, Ludwig Schmidt, Dimitris Tsipras, and Adrian Vladu. Towards deep learning models resistant to adversarial attacks. In ICLR, 2018.

[2] Croce, Francesco, and Matthias Hein. "Reliable evaluation of adversarial robustness with an ensemble of diverse parameter-free attacks." In ICML, 2020.

[3] Andriushchenko, Maksym, et al. "Square attack: a query-efficient black-box adversarial attack via random search." In ECCV, 2020.

[4] Rauber, Jonas, et al. "Foolbox native: Fast adversarial attacks to benchmark the robustness of machine learning models in pytorch, tensorflow, and jax." Journal of Open Source Software 5.53 (2020): 2607.

[5] Duan, Ranjie, et al. "Adversarial laser beam: Effective physical-world attack to dnns in a blink." In CVPR. 2021.

---

### Decision · Program_Chairs · 2023-01-20

**Decision:**

Accept: notable-top-25%

**Justification For Why Not Higher Score:**

The work is limited on the MDE task with self-supervised learning, but in practice, the depth estimation models may be trained with supervised learning.

**Justification For Why Not Lower Score:**

The overall method is interesting and the method is effective.

**Metareview: Summary, Strengths And Weaknesses:**

This paper introduces an adversarial training scheme for self-supervised monocular depth estimation. The work employs the L0-norm-bounded perturbation during training for enhancing the adversarial robustness against physical-world attacks. The authors show that this method is effective in experiments. Overall, the idea and the framework seems to be effective and novel, and all the reviewers are positive on this paper. The main weakness is that more experiments are needed, and the clarity can be further improved, which yet can be addressed in the final version. Authors need to carefully consider the comments of all reviewers, and improve the final version accordingly.

**Note From Pc:**

if the above contains the word "oral" or "spotlight" please see: "oral" presentation means -> notable-top-5% and "spotlight" means -> notable-top-25%. As stated in our emails, we are disassociating presentation type from AC recommendations